

# Improving the EnSRF in the Community Inversion Framework: a case study with ICON-ART 2024.01

Joël Thanwerdas[1], Antoine Berchet[2], Lionel Constantin[1], Aki Tsuruta[3], Michael Steiner[1], Friedemann Reum[4], Stephan Henne[1], and Dominik Brunner[1]

[1]Empa, Swiss Federal Laboratories for Materials Science and Technology, Dübendorf, Switzerland
[2]Laboratoire des Sciences du Climat et de l'Environnement, CEA-CNRS-UVSQ, Gif-sur-Yvette, France
[3]Finnish Meteorological Institute (FMI), Helsinki, Finland
[4]German Aerospace Center (DLR), Institute of Atmospheric Physics, Oberpfaffenhofen, Germany

**Correspondence:** Joël Thanwerdas (joel.thanwerdas@empa.ch)

**Abstract.** The Community Inversion Framework (CIF) brings together methods for estimating greenhouse gas fluxes from atmospheric observations. While the analytical and variational optimization methods implemented in CIF are operational and have proved to be accurate and efficient, the initial ensemble method was found to be incomplete and could hardly be compared to other ensemble methods employed in the inversion community, mainly owing to strong performance limitations and absence

of localization methods. In this paper, we present and evaluate a more efficient implementation of the ensemble mode. As a first step, we chose to implement the serial and batch versions of the Ensemble Square Root Filter (EnSRF) algorithm because it is widely employed in the inversion community. We provide a comprehensive description of the technical implementation in CIF and the useful features it can provide to users. Finally, we demonstrate the capabilities of the CIF-EnSRF system using a large number of synthetic experiments over Europe, exploring the system's sensitivity to multiple parameters that can be tuned

by users. As expected, the results are sensitive to the ensemble size and localization parameters. Other tested parameters, such as the number of lags, the propagation factors, or the localization function can also have a substantial influence on the results. We also introduce and provide a way of interpreting a set of metrics that are automatically computed by CIF and that can help assessing the success of inversions and comparing them. This work complements previous efforts focused on other inversion methods within CIF. With the integration of these new ensemble algorithms, any chemical transport model (CTM), including

models without existing adjoint, can now perform inversions using CIF, leveraging its robust capabilities.

## 1 Introduction

Global warming is caused by the accumulation of greenhouse gases (GHGs) in the atmosphere such as carbon dioxide ($CO_2$), methane ($CH_4$), nitrous oxide ($N_2O$) or synthetic gases. The atmospheric concentrations of these GHGs have been drastically increasing since the pre-industrial era (in 2019 compared to 1750, $CO_2$: +47 %; $CH_4$: +156 %; $N_2O$: +23 %; Gulev et al.,

2021), due to the intensification of human activities worldwide. As the international community recognized the existence of the link between human activities and global warming, the urge to gain a comprehensive understanding of the varied sources of GHGs, both natural and anthropogenic, across diverse sectors and geographical regions, has been intensifying.



In response to this imperative, concerted efforts have been made to continuously develop observational networks across the globe (e.g., Schuldt et al., 2023; Ramonet et al., 2020; Prinn et al., 2000; Dlugokencky et al., 1994). In tandem with the ground-based networks, advancements in remote sensing technologies have considerably expanded geographical coverage and enabled frequent observations over remote areas (e.g., Taylor et al., 2023; Lauvaux et al., 2022; Suto et al., 2021; Parker et al., 2020; Hu et al., 2018; Frankenberg et al., 2006). These ever-growing datasets generated by monitoring networks and satellite observations provide an unprecedented wealth of information on greenhouse gases and call for innovative techniques, such as data assimilation methods, capable of extracting pertinent information from these data.

Data assimilation methods have been originally designed for Numerical Weather Prediction (NWP) to deal with the chaotic behavior of the atmosphere (Ghil and Malanotte-Rizzoli, 1991). Data assimilation allows to integrate observational information into complex NWP models and continuously refine and update their predictions, therefore providing better analysis and forecast of the atmospheric state. Given the established efficacy of data assimilation techniques in weather forecasting, they found a natural extension into the realm of GHG flux estimation in the late 1980s and early 1990s (Enting and Newsam, 1990a; Newsam and Enting, 1988). For these applications, the term "inversion" is preferred to "data assimilation". The explanation is simple: a chemical transport model (CTM), serves as an operator linking input data (e.g., fluxes) and observable quantities (e.g., atmospheric concentrations). The input data are only boundary conditions for the prognostic equations solved by the model to obtain a numerical estimate of the observable quantities. When observations of these quantities are utilized to refine model input, the process is said to be inverted.

Over time, multiple inversion methods have been designed by the scientific community to provide optimized estimates of fluxes. Despite important differences between these algorithms, they share a common theoretical foundation, which is Bayes' theorem, and aim to minimize a specific cost function. These algorithms can be broadly classified into four distinct groups: analytical (e.g., Wittig et al., 2023; Wang et al., 2018; Bousquet et al., 2011; Stohl et al., 2009; Kopacz et al., 2009), variational (e.g., Thanwerdas et al., 2024; Fortems-Cheiney et al., 2021; Chevallier et al., 2010, 2005), ensemble (e.g., Steiner et al., 2024; Tsuruta et al., 2017; van der Laan-Luijkx et al., 2015; Bruhwiler et al., 2014; Kim et al., 2014a, b; Peters et al., 2007, 2005), and Monte Carlo Markov Chain (MCMC) methods (e.g., Zammit-Mangion et al., 2016; Miller et al., 2014; Ganesan et al., 2014; Mukherjee et al., 2011), each presenting a particular set of strengths and weaknesses. Within the inversion community, individual research groups commonly designed and employed distinct combinations of inversion systems and CTMs with varying transparency of specific implementations and their continuous development, applying them to a range of trace gases and various types of observations depending on the application. This variety of combinations, coupled with the lack of transparency regarding the advancements, poses a challenge to the inversion community in terms of leveraging previous developments and avoiding redundant feature development.

The Community Inversion Framework (CIF, Berchet et al., 2021, or hereinafter BA21) has been designed to bring together the different inversion methods and CTMs used in the community. It is built as an open source, thoroughly documented, highly modular and multi-model inversion framework written in Python, that facilitates the comparison of 1) inversion methods and 2) CTMs. Additionally, CIF is being constantly updated and enhanced, based on user feedback. Consequently, it serves as a robust foundation upon which the community can build and continue to produce accurate estimates of GHG (and other species)



fluxes in a reasonable computational time. It is important to note that other similar inversion systems exist and are used in the
inversion community. One prominent example is the Carbon Tracker Data Assimilation Shell (CTDAS, van der Laan-Luijkx
et al., 2017; Peters et al., 2005), a well-established system widely employed for deriving optimized estimates of GHG fluxes,
mainly with ensemble methods (Steiner et al., 2024; Tsuruta et al., 2023; He et al., 2018).

The analytical and variational methods implemented in CIF are operational and have proved to be accurate and computation-
ally performant (Fortems-Cheiney et al., 2023; Wittig et al., 2023; Savas et al., 2023; Remaud et al., 2022; Thanwerdas et al.,
2024, 2022a, b). However, analytical methods become excessively expensive for large inversion problems and CTMs without
adjoint cannot use the variational methods. The increasing need for running CIF inversions with these CTMs has therefore
made the imperative to employ efficient ensemble methods more pressing. However, the initial ensemble method presented in
BA21 was found to be incomplete and could hardly be compared to other published ensemble methods, mainly owing to strong
performance limitations, absence of localization methods, but also to errors in the generation of ensembles and the propagation
of information from one cycle to the other. This method therefore needed improvements.

This work presents the recent enhancements to the ensemble method in CIF. Section 2 introduces the conceptual framework
governing ensemble methods, with a specific focus on the method implemented in CIF. Section 3 describes the technical
implementation of this method and highlights the main benefits for the inversion community. Section 4 demonstrates the
potential of this new method using a large set of experiments with synthetic data. Section 5 provides a summary of the key
findings and explores envisioned future developments.

## 2 Theoretical formulation

Here, we provide an overview of the general theoretical framework designed for atmospheric inversions (Enting, 2002; Enting
et al., 1995, 1993; Enting and Newsam, 1990b; Tarantola, 1987; Cunnold et al., 1983; Gelb, 1974), with a specific focus on
ensemble methods.

### 2.1 Kalman Filter

An atmospheric inversion seeks to optimize the variables included in the *control vector* (also called state or target vector),
denoted by $\boldsymbol{x}$ [of dimension $n$], based on the *observation vector* $\boldsymbol{y}^{\mathrm{o}}$ [$p$]. An *observation operator* $\mathcal{H}(.)$ [$n \mapsto p$] links the
control space and the observation space, where the control and observation vectors are respectively defined. In the Bayesian
approach, a *prior* (or background) *control vector* $\boldsymbol{x}^{\mathrm{b}}$ is updated such that the resulting *posterior* (or analysis) *control vector* $\boldsymbol{x}^{\mathrm{a}}$
maximizes the conditional probability density $p(\boldsymbol{x}|\boldsymbol{y}^{\mathrm{o}})$. Bayes' theorem states that,

$$p(\boldsymbol{x}|\boldsymbol{y}^{\mathrm{o}}) \propto p(\boldsymbol{y}^{\mathrm{o}}|\boldsymbol{x}) \cdot p(\boldsymbol{x}) \tag{1}$$

Errors in the observations and the prior control vector in atmospheric inversions are typically assumed to be unbiased,
although it is difficult to accurately characterize potential biases. Gaussian distributions, denoted by $\mathcal{N}(.)$, are frequently as-
sumed to represent errors for two main reasons. 1) Errors can be thought of as the sum of several small, independent effects (i.e.





random variables). According to the central limit theorem, this sum tends to follow a Gaussian distribution. Consequently, as-
suming such a distribution is reasonable in the absence of better information. 2) Algorithms that assume Gaussian distributions
are generally simpler to understand and implement because this assumption simplifies the mathematics involved. Consequently,
the probability density functions associated to the errors are defined by,

$$p(\boldsymbol{x}) = \mathcal{N}(\boldsymbol{x}^{\mathrm{b}}, \mathbf{B}) \Rightarrow p(\boldsymbol{x} - \boldsymbol{x}^{\mathrm{b}}) = p(\boldsymbol{\epsilon}^{\mathrm{b}}) = \mathcal{N}(\mathbf{0}, \mathbf{B}) \tag{2}$$

$$p(\boldsymbol{y}^{\mathrm{o}}|\boldsymbol{x}) = \mathcal{N}(\mathcal{H}(\boldsymbol{x}), \mathbf{R}) \Rightarrow p(\boldsymbol{y}^{\mathrm{o}} - \mathcal{H}(\boldsymbol{x})|\boldsymbol{x}) = p(\boldsymbol{\epsilon}^{\mathrm{o}}|\boldsymbol{x}) = \mathcal{N}(\mathbf{0}, \mathbf{R}) \tag{3}$$

$\boldsymbol{\epsilon}^{\mathrm{b}} = \boldsymbol{x} - \boldsymbol{x}^{\mathrm{b}}$ and $\boldsymbol{\epsilon}^{\mathrm{o}} = \boldsymbol{y}^{\mathrm{o}} - \mathcal{H}(\boldsymbol{x})$ are the *background* and *observation errors*, respectively. $\mathbf{B} = \mathbb{E}[(\boldsymbol{\epsilon}^{\mathrm{b}})(\boldsymbol{\epsilon}^{\mathrm{b}})^{\mathrm{T}}]$ and $\mathbf{R} = \mathbb{E}[(\boldsymbol{\epsilon}^{\mathrm{o}})(\boldsymbol{\epsilon}^{\mathrm{o}})^{\mathrm{T}}]$ are the *background-error* and *observation-error covariance matrices*, respectively, with $\mathbb{E}[.]$ the expectation op-
erator. When the probability distributions $p(\boldsymbol{x})$ and $p(\boldsymbol{y}^{\mathrm{o}}|\boldsymbol{x})$ are Gaussian, the left hand side of Bayes' theorem in Eq.( 1) also
follows a Gaussian distribution,

$$p(\boldsymbol{x}|\boldsymbol{y}^{\mathrm{o}}) = \mathcal{N}(\boldsymbol{x}^{\mathrm{a}}, \mathbf{P}^{\mathrm{a}}) \Rightarrow p(\boldsymbol{x} - \boldsymbol{x}^{\mathrm{a}}|\boldsymbol{y}^{\mathrm{o}}) = p(\boldsymbol{\epsilon}^{\mathrm{a}}|\boldsymbol{y}^{\mathrm{o}}) = \mathcal{N}(\mathbf{0}, \mathbf{P}^{\mathrm{a}}) \tag{4}$$

$\boldsymbol{\epsilon}^{\mathrm{a}} = \boldsymbol{x} - \boldsymbol{x}^{\mathrm{a}}$ and $\mathbf{P}^{\mathrm{a}} = \mathbb{E}[(\boldsymbol{\epsilon}^{\mathrm{a}})(\boldsymbol{\epsilon}^{\mathrm{a}})^{\mathrm{T}}]$ define the *analysis error and analysis-error covariance matrix*, respectively. It follows
that $\boldsymbol{x}^{\mathrm{a}}$ is the vector minimizing the quadratic cost function $J(.)$ defined by,

$$J(\boldsymbol{x}) = \frac{1}{2}(\boldsymbol{x} - \boldsymbol{x}^{\mathrm{b}})^{\mathrm{T}}\mathbf{B}^{-1}(\boldsymbol{x} - \boldsymbol{x}^{\mathrm{b}}) + \frac{1}{2}(\mathcal{H}(\boldsymbol{x}) - \boldsymbol{y}^{\mathrm{o}})^{T}\mathbf{R}^{-1}(\mathcal{H}(\boldsymbol{x}) - \boldsymbol{y}^{\mathrm{o}}) \tag{5}$$

$$= J^{\mathrm{b}}(\boldsymbol{x}) + J^{\mathrm{o}}(\boldsymbol{x}) \tag{6}$$

$J^{\mathrm{o}}(\boldsymbol{x})$ and $J^{\mathrm{b}}(\boldsymbol{x})$ are the contributions of the observations and the background to the total cost function, respectively. Mini-
mizing $J$ means finding the optimal balance between fitting the atmospheric measurements and remaining close to the prior
estimate. The error covariance matrices determine the relative weight assigned to each of these objectives. If it is additionally
assumed that $\mathcal{H}$ is linear, $\mathcal{H}(\boldsymbol{x}) = \mathbf{H}\boldsymbol{x}$, where $\mathbf{H}$ is the Jacobian matrix of $\mathcal{H}$, the analytical solution for $\boldsymbol{x}^{\mathrm{a}}$ is given by

$$\boldsymbol{x}^{\mathrm{a}} = \boldsymbol{x}^{\mathrm{b}} + \mathbf{K}(\boldsymbol{y}^{\mathrm{o}} - \mathbf{H}\boldsymbol{x}^{\mathrm{b}}) \tag{7}$$

with

$$\mathbf{K} = \mathbf{B}\mathbf{H}^{\mathrm{T}}(\mathbf{H}\mathbf{B}\mathbf{H}^{\mathrm{T}} + \mathbf{R})^{-1} \tag{8}$$

$\mathbf{K}\ [n \times p]$ is called *the gain matrix*.

Utilizing the Sherman–Morrison–Woodbury identity, it can also be expressed as,

$$\mathbf{K} = (\mathbf{B}^{-1} + \mathbf{H}^{\mathrm{T}}\mathbf{R}^{-1}\mathbf{H})^{-1}\mathbf{H}^{\mathrm{T}}\mathbf{R}^{-1} \tag{9}$$

Using Eq. (7) and (8), it is also possible to derive an analytical formulation for the analysis-error covariance matrix,

$$\mathbf{P}^{\mathrm{a}} = (\mathbf{I}_{\mathbf{n}} - \mathbf{K}\mathbf{H})\mathbf{B} \tag{10}$$





where $\mathbf{I_n}$ is the identity matrix of size $n$.

This analytical solution in Eq. (8) is the update phase of the so-called Kalman filter (KF; Kalman, 1960) which was specifically designed to optimize a prior estimate of a state vector using a set of observations. Other teams have extended this framework to non-Gaussian distributions (e.g., truncated Gaussian densities, semiexponential, log-normal distributions, etc.;

Lunt et al., 2016; Miller et al., 2014; Ganesan et al., 2014; Bergamaschi et al., 2010; Michalak and Kitanidis, 2005), albeit this complicates the derivation of the solution. Additionally, an alternative version of the KF, known as the Extended Kalman Filter (EKF, Evensen, 1993, 1992; Brunner et al., 2012), can be employed when $\mathcal{H}$ is non-linear.

There are subtle differences in the utilization of KF equations between NWP and inversion applications. In the context of NWP, optimization of the control vector occurs at different timesteps (analysis steps), incorporating both the prior control

vector and the observations available at that timestep. After the analysis at timestep $t$, a forecast operator, denoted by $\mathcal{F}$, uses the newly optimized state to advance the prediction and generates the background control vector for the following assimilation timestep $t+1$,

$$\boldsymbol{x}_{t+1}^{\mathrm{b}} = \mathcal{F}(\boldsymbol{x}_t^{\mathrm{a}}). \tag{11}$$

Equation (11) describes the evolution of meteorological fields due to complex nonlinear dynamical processes linking two

different timesteps. In the context of atmospheric inversion, in contrast, the forecast model links the optimized fluxes at one assimilation timestep to the background fluxes at the next assimilation timestep. Since no established relationship exists, persistence of fluxes is often assumed for simplicity (Brunner et al., 2012; Peters et al., 2005). Additionally, deriving the observation operator $\mathbf{H}$ in matrix format is more challenging in the context of inversion than in NWP. This is because defining the relationship between the control space (e.g., fluxes) and the observation space (e.g., atmospheric concentrations) requires a CTM,

whereas in NWP, control variables are often observed.

The analytical inversion method directly applies the KF equations presented above to derive the optimal solution. However, the explicit derivation of $\mathbf{H}$ and its transpose $\mathbf{H}^{\mathrm{T}}$ requires $n$ forward runs of the CTM, or $\min(n, p)$ forward runs in case the CTM is Lagrangian or if an adjoint version of the model is available. Building $\mathbf{H}$ can therefore be excessively expensive, especially when both optimizing numerous variables and assimilating a large number of observations.

The other two methods, variational and ensemble, offer different solutions to cope with this limitation. In this study, we focus on ensemble methods.

## 2.2 Ensemble Kalman Filter and square root filters

The ensemble methods utilized in atmospheric inversions drew inspiration from the original Ensemble Kalman Filter (EnKF) introduced by Evensen (1994) for NWP. EnKF, rooted in Monte Carlo methods, was initially designed to surpass the results

of the EKF, avoid the linearization of a nonlinear forecast model and enhance the derivation of forecast-error statistics after each analysis. The principle is that an ensemble of vectors is used to represent the probability distribution of the control vector. Each member of the ensemble produces a different forecast, and the ensembles of control vectors and forecasts are used to compute the posterior control vector $\boldsymbol{x}^{\mathrm{a}}$ using Eq. 7. This algorithm has undergone improvements through subsequent studies



(Houtekamer and Mitchell, 1998; Burgers et al., 1998). In particular, these studies account for measurement noise by creating
a perturbed observation vector for each member of the ensemble. This enhanced algorithm is now recognized as the stochastic
EnKF.

A few years later, deterministic versions of the EnKF were developed: Ensemble Transform Kalman Filter (ETKF; Bishop
et al., 2001), Ensemble Adjustment Kalman filter (EAKF; Anderson, 2001), Ensemble Square Root Filter (EnSRF;  Whitaker
and Hamill, 2002) and Local Ensemble Transform Kalman Filter (LETKF; Hunt et al., 2007), to circumvent sampling issues
associated with the use of perturbed observations. Deterministic methods have been shown to be more accurate than their
stochastic counterparts (e.g., Tippett et al., 2003). It should be emphasized that despite the name chosen for the EnSRF, all the
aforementioned deterministic versions of EnKF belong to the family of square root filters (Livings et al., 2008; Tippett et al.,
2003).

In a square root filter, the background-error covariance matrix is decomposed as $\mathbf{B} = \mathbf{Z}\mathbf{Z}^\mathrm{T}$ where $\mathbf{Z}$ is a square root matrix
of $\mathbf{B}$ $[n \times N]$. In an ensemble representation, $N$ denotes the number of samples in the ensemble and we further define $\mathbf{X}$ and
$\mathbf{X}'$ such that,

$$\mathbf{Z} = \frac{1}{\sqrt{N-1}}(\mathbf{X} - \overline{\boldsymbol{x}}\mathbb{1}) = \frac{1}{\sqrt{N-1}}\mathbf{X}' \tag{12}$$

where $\mathbb{1} = (1,...,1)$ is the unit matrix of dimension $[1 \times N]$, $\mathbf{X} = (\boldsymbol{x_1},...,\boldsymbol{x_N})$ $[n \times N]$ represents an ensemble of $N$ control
vectors and $\mathbf{X}' = (\boldsymbol{x'_1},...,\boldsymbol{x'_N})$ $[n \times N]$ represents the deviations around the mean $\overline{x} = \frac{1}{N}\sum_{i=1}^{N} \boldsymbol{x_i}$ $[n \times 1]$.

This definition of the square root of $\mathbf{B}$ offers an intuitive approach to solving the inversion problem: we create an ensemble of
perturbed control vectors $\boldsymbol{x_i}$ that samples the prior distribution $\mathcal{N}(\boldsymbol{x}^\mathrm{b}, \mathbf{B})$, and then we employ the KF equations to incorporate
observational knowledge and approximate the posterior distribution $\mathcal{N}(\boldsymbol{x}^\mathrm{a}, \mathbf{P}^\mathrm{a})$. In the limit of $N \to \infty$, the covariance matrix
calculated from $\mathbf{X}'$ is equal to $\mathbf{B}$. However, in a practical scenario where $N$ is relatively small compared to $n$ and the dimension
of $\mathbf{B}$, we can only achieve an approximation of $\mathbf{B}$, denoted by $\mathbf{B_N}$,

$$\mathbf{B_N} = \frac{1}{N-1}\mathbf{X}'\mathbf{X}'^\mathrm{T} \xrightarrow[N\to\infty]{} \mathbf{B} \tag{13}$$

The primary benefit of the ensemble method is the ability to approximate the *model-observation covariance* $\mathbf{B}\mathbf{H}^\mathrm{T}$ and the
*observation-observation covariance* $\mathbf{H}\mathbf{B}\mathbf{H}^\mathrm{T}$ in Eq. (8) without the necessity of explicitly computing $\mathbf{H}$,

$$\mathbf{B}\mathbf{H}^\mathrm{T} \approx \mathbf{B_N}\mathbf{H}^\mathrm{T} = \frac{1}{N-1}\mathbf{X}'\mathbf{Y}'^\mathrm{T} \tag{14}$$

$$\mathbf{H}\mathbf{B}\mathbf{H}^\mathrm{T} \approx \mathbf{H}\mathbf{B_N}\mathbf{H}^\mathrm{T} = \frac{1}{N-1}\mathbf{Y}'\mathbf{Y}'^\mathrm{T} \tag{15}$$

where $\mathbf{Y}' = \mathbf{H}\mathbf{X}' = (\mathbf{H}\boldsymbol{x'_1},...,\mathbf{H}\boldsymbol{x'_N}) = (\boldsymbol{y'_1},...,\boldsymbol{y'_N})$.

Consequently, the columns of $\mathbf{Y}'$ $[p \times N]$ are obtained by transporting $N+1$ sample tracers with the CTM: one tracer
associated with the ensemble mean $\overline{x}$ and $N$ tracers associated with the deviations $\boldsymbol{x_i}'$. The perturbed control vectors $\boldsymbol{x_i}$ can
also be transported instead of the deviations because $\mathbf{H}\boldsymbol{x'_i} = \mathbf{H}\boldsymbol{x_i} - \mathbf{H}\overline{\boldsymbol{x}}$. In CIF, this option is preferred.

The Kalman gain matrix $\mathbf{K}$ can be explicitly computed and the ensemble mean is updated using Eq. (7),

$$\overline{\boldsymbol{x}}^\mathrm{a} = \overline{\boldsymbol{x}} + \frac{1}{N-1}\mathbf{X}'\mathbf{Y}'^\mathrm{T}\mathbf{D}^{-1}\boldsymbol{d} \tag{16}$$





where $\boldsymbol{d} = \boldsymbol{y}^{\mathrm{o}} - \mathbf{H}(\overline{\boldsymbol{x}})$ is the *innovation vector* and $\mathbf{D} = \frac{1}{N-1}\mathbf{Y}'\mathbf{Y}'^{\mathrm{T}} + \mathbf{R}$ is the *innovation covariance matrix*.

The analysis ensemble is then given by,

$$\mathbf{X}^{\mathrm{a}} = \overline{\boldsymbol{x}}^{\mathrm{a}}\mathbb{1} + \mathbf{X}'^{\mathrm{a}} \tag{17}$$

where the updated deviations $\mathbf{X}'^{\mathrm{a}}$ cannot be simply calculated using an equivalent of Eq. (16). In a deterministic EnKF algo-
rithm, the analysis-error covariance matrix is formed using the square root formulation,

$$\mathbf{P}^{\mathrm{a}} = \frac{1}{N-1}(\mathbf{X}'^{\mathrm{a}})(\mathbf{X}'^{\mathrm{a}})^{\mathrm{T}} \tag{18}$$

It must approximate its Kalman filter counterpart,

$$\mathbf{P}^{\mathrm{a}} = (\mathbf{I_n} - \mathbf{KH})\mathbf{B_N} \tag{19}$$

$$= \frac{1}{N-1}(\mathbf{I_n} - \mathbf{KH})\mathbf{X}'\mathbf{X}'^{\mathrm{T}} \tag{20}$$

$$= \frac{1}{N-1}\mathbf{X}'(\mathbf{I_N} - \mathbf{Y}'^{\mathrm{T}}\mathbf{D}^{-1}\mathbf{Y}')\mathbf{X}'^{\mathrm{T}} \tag{21}$$

It follows that,

$$\mathbf{X}'^{\mathrm{a}} = \mathbf{X}'\mathbf{T} \tag{22}$$

where $\mathbf{T}\ [N \times N]$ is called the *transform matrix* and satisfies,

$$\mathbf{T}\mathbf{T}^{\mathrm{T}} = \mathbf{I_N} - \mathbf{Y}'^{\mathrm{T}}\mathbf{D}^{-1}\mathbf{Y}' \tag{23}$$

The solution of this equation is not unique because if we define $\mathbf{L} = \mathbf{T}\mathbf{U}$ where $\mathbf{U}$ is any orthogonal transform $\mathbf{U}\mathbf{U}^{\mathrm{T}} = \mathbf{U}^{\mathrm{T}}\mathbf{U} = \mathbf{I_N}$, then $\mathbf{L}$ is also a solution. Hence, the definition of $\mathbf{T}$ is a key difference between the deterministic algorithms. In the next section, we will focus on the algorithm we chose to implement in CIF.

### 2.3 Ensemble Square Root Filter

The EnSRF was already employed with the models TM5 (Krol et al., 2005), WRF-Chem (Skamarock et al., 2021; Grell et al., 2005), STILT (Lin et al., 2003) and ICON-ART (Schröter et al., 2018; Zängl et al., 2015) to perform inversions for different species and at different scales (Steiner et al., 2024; Reum, 2024; Mannisenaho et al., 2023; Tsuruta et al., 2023; He et al., 2018; Tsuruta et al., 2017). Hence, to foster interest from other inverse modelling groups and to allow them to directly compare with their existing tools, BA21 implemented a preliminary version of the EnSRF in CIF as a first step. We elaborate on this method in detail in this section.

#### 2.3.1 Batch EnSRF

In Whitaker and Hamill (2002), the authors investigated a formulation in which

$$\mathbf{X}'^{\mathrm{a}} = (\mathbf{I_n} - \hat{\mathbf{K}}\mathbf{H})\mathbf{X}' \tag{24}$$



where $\hat{\mathbf{K}}$ is sought such that Eq. (10) is satisfied. The solution is

$$\hat{\mathbf{K}} = \mathbf{B_N}\mathbf{H^T}\mathbf{D}^{-\frac{1}{2}}(\mathbf{D}^{\frac{1}{2}} + \mathbf{R}^{\frac{1}{2}})^{-1} = \frac{1}{N-1}\mathbf{X'}\mathbf{Y'^T}\mathbf{D}^{-\frac{1}{2}}(\mathbf{D}^{\frac{1}{2}} + \mathbf{R}^{\frac{1}{2}})^{-1} \tag{25}$$

As the derivation is not trivial and can be found in Whitaker and Hamill (2002) and references therein, we refrain from presenting it here. It follows that,

$$\mathbf{X'^a} = (\mathbf{I_n} - \hat{\mathbf{K}}\mathbf{H})\mathbf{X'} = \mathbf{X'}(\mathbf{I_N} - \frac{1}{N-1}\mathbf{Y'^T}\mathbf{V}\mathbf{Y'}) \tag{26}$$

where $\mathbf{V} = \mathbf{D}^{-\frac{1}{2}}(\mathbf{D}^{\frac{1}{2}} + \mathbf{R}^{\frac{1}{2}})^{-1}$ Note that this formulation also defines the transformation matrix $\mathbf{T} = \mathbf{I_N} - \frac{1}{N-1}\mathbf{Y'^T}\mathbf{V}\mathbf{Y'}$ for the EnSRF. Since this version of the EnSRF assimilates the observations simultaneously, it is referred to as the batch EnSRF.

### 2.3.2  Serial EnSRF

Whitaker and Hamill (2002) also introduced an alternative approach, called the serial EnSRF. In the serial EnSRF algorithm, the observations are processed serially (one at a time), in order to reduce the substantial computational cost that can be associated with matrix inversion. This is feasible only when observation errors are uncorrelated, namely when the $\mathbf{R}$ matrix is diagonal. When the single observation $j$ is assimilated, $\mathbf{R}$, $\mathbf{D}$ and $\boldsymbol{d}$ become scalars, denoted by $r_j$, $D_j$ and $d_j$. Additionally, $\mathbf{Y'}$ is

reduced to an $N$-dimensional vector, denoted by $\boldsymbol{y'_j}$.

Consequently, Eq.( 16) and Eq.( 26) are revised as follows to update the mean and deviations of the ensemble based on observation $j$:

$$\overline{\boldsymbol{x}}^a = \overline{\boldsymbol{x}} + d_j\boldsymbol{k_j} \tag{27}$$

$$\mathbf{X'^a} = \mathbf{X'} - \alpha\boldsymbol{k_j}\boldsymbol{y'_j}^T \tag{28}$$

where $\boldsymbol{k_j} = \frac{1}{D_j}\frac{1}{N-1}\mathbf{X'}\boldsymbol{y'_j}$ and $\alpha = (1 + \sqrt{\frac{r_j}{D_j}})^{-1}$. After each observation is assimilated, the analyzed state is used as the new background for the next observation, until all observations are processed. Consequently, the vector $\mathbf{Y'}$ must also be updated at each step. As the observation operator is assumed to be linear, it is calculated as,

$$\overline{\boldsymbol{y}}^a = \mathbf{H}\overline{\boldsymbol{x}}^a = \mathbf{H}\overline{\boldsymbol{x}} + d_j\boldsymbol{l_j} \tag{29}$$

$$\mathbf{Y'^a} = \mathbf{Y'} - \alpha\boldsymbol{l_j}\boldsymbol{y'_j}^T \tag{30}$$

where $\boldsymbol{l_j} = \frac{1}{D_j}\frac{1}{N-1}\mathbf{Y'}\boldsymbol{y'_j}$

All observations are processed until the final analyzed state is reached. Note that the batch EnSRF and serial EnSRF are mathematically equivalent if observation errors are uncorrelated (Kotsuki et al., 2017; Nerger, 2015; Whitaker and Hamill, 2002) and thus provide the same results.

### 2.3.3  Ensemble Square Root Smoother

After the KF theory presented in Sect. 2.1 had been applied in several studies to estimate surface emissions of trace gases (e.g., Haas-Laursen et al., 1996; Hartley and Prinn, 1993), Bruhwiler et al. (2005) introduced the Fixed Lag Kalman Smoother to



reduce the computational cost associated with the processing of a large number of observations. The authors initially observed that due to atmospheric mixing, information from a specific source location does not propagate to atmospheric concentrations very far into the future. As a result, only a subset of observations obtained after the emission, around the location of the source, is necessary to effectively constrain past fluxes. The time period over which transport information is retained is called the fixed-lag and depends on the scale of the application (e.g., several months for the global scale but less for the regional scale).

Peters et al. (2005) integrated this fixed-lag feature from Bruhwiler et al. (2005) with the serial EnSRF algorithm from Whitaker and Hamill (2002), which was later developed further into CTDAS (van der Laan-Luijkx et al., 2017). In this system, the full assimilation period is split into windows of finite length. For each window, fluxes within the window are optimized using both the observations from the current window and those from a fixed number (lag) of subsequent windows. This version of the EnSRF algorithm, which is the focus of this work, is described in detail in Sect. 3.1. It is worth noting that while Peters et al. (2005) retained the name EnSRF, their method could also be referred to as the Ensemble Square Root Smoother (EnSRS).

### 2.3.4 Covariance localization

Due to sampling errors, spurious long-range correlations tend to appear in $\mathbf{B_N}$, which can ultimately lead to a degraded analysis. The so-called covariance localization technique has been developed to mitigate this effect by filtering out the correlations between distant locations or between variables that have small correlations (Hamill et al., 2001; Houtekamer and Mitchell, 2001, 1998).

Localization is typically performed by applying a Schur product (element-wise multiplication, denoted by $\circ$) between a covariance matrix and a localization matrix $\mathbf{L}$ $[n \times p]$. Each element $\mathbf{L}_{i,j}$ is defined using some decreasing function of the distance between the locations of the $i$-th and $j$-th elements. Two types of localization exist: while the $\mathbf{R}$-localization is applied on the observation error covariance matrix $\mathbf{R}$, the $\mathbf{B}$-localization operates on the background-error covariance matrix $\mathbf{B}$ (Hotta and Ota, 2021).

The $\mathbf{B}$-localization can be further split into the model-space localization and the observation-space localization (Shlyaeva et al., 2019). The model-space $\mathbf{B}$-localization directly transforms $\mathbf{B_N} = \frac{1}{N-1}\mathbf{X}'\mathbf{X}'^{\mathrm{T}}$ and the gain matrix $\mathbf{K}$. When applied to the batch EnSRF equations, we have

$$\mathbf{K_{loc,model}} = (\mathbf{L} \circ \mathbf{B_N})\mathbf{H}^{\mathrm{T}}(\mathbf{H}(\mathbf{L} \circ \mathbf{B_N})\mathbf{H}^{\mathrm{T}} + \mathbf{R})^{-1} \tag{31}$$

The observation-space $\mathbf{B}$-localization modifies separately the model-observation covariance $\mathbf{B_N}\mathbf{H}^{\mathrm{T}} = \frac{1}{N-1}\mathbf{X}'\mathbf{Y}'^{\mathrm{T}}$ and the observation-observation covariance $\mathbf{H}\mathbf{B_N}\mathbf{H}^{\mathrm{T}} = \frac{1}{N-1}\mathbf{Y}'\mathbf{Y}'^{\mathrm{T}}$ . Two different localization matrices $\mathbf{L_1}$ $[n \times p]$ and $\mathbf{L_2}$ $[p \times p]$ are therefore necessary,

$$\mathbf{K_{loc,obs}} = (\mathbf{L_1} \circ (\mathbf{B_N}\mathbf{H}^{\mathrm{T}}))(\mathbf{L_2} \circ (\mathbf{H}\mathbf{B_N}\mathbf{H}^{\mathrm{T}}) + \mathbf{R})^{-1} \tag{32}$$

In the context of inversion performed with EnSRF, observation-space $\mathbf{B}$-localization is preferred over model-space $\mathbf{B}$-localization because $\mathbf{H}$ is not explicitly computed. It is important to note that applying localization invalidates the mathematical equivalence between serial and batch EnSRF, as well as between serial EnSRF algorithms executed with different assimilation orders (Kotsuki et al., 2017; Nerger, 2015).



## 3 Technical implementation of the CIF-EnSRF

Many improvements have been introduced since the initial implementation of EnSRF in CIF by BA21. Here, we describe the new CIF-EnSRF workflow comprehensively and highlight the various enhancements.

### 3.1 Implementation details

The objective of an inversion performed with the EnSRF method is to optimize elements within a control vector, encompassing fluxes, boundary conditions, atmospheric concentrations, and potentially more. In our demonstration, we specifically focus on fluxes and optimize scaling factors applied to a prior estimate of these fluxes. The full assimilation time period is partitioned into several windows. For each window, the time duration and the number of scaling factors to optimize are identical. Consequently, selecting a shorter window results in higher temporal resolution for the optimized scaling factors. However, if the number of lags is unchanged, a shorter window also means that the influence of the scaled fluxes propagates to assimilated observations that are closer to the sources, as each cycle is also shorter. One of the challenges in this inversion process is effectively managing the trade-off between the window length, the number of lags, and the computational cost.

#### 3.1.1 Initialization and generation of samples

Through the configuration file of CIF, users can define fundamental settings for the inversion process:

- **datei**: Start date of the inversion.

- **datef**: End date of the inversion.

- **window_length**: Length of a single window.

- **nlag**: Number of windows within each cycle. Consequently, it also represents the number of times the control variables within a window are optimized by the system.

As an illustrative example, we consider an inversion with the following settings:

- **datei**: 2018-01-01

- **datef**: 2018-03-02

- **window_length**: 10D (i.e., ten days)

- **nlag**: 2

With these settings, the resulting inversion is composed of six cycles of 20 days each (two windows of 10 days). When an inversion is started with CIF, the system first reads the configuration file and initializes all the relevant components, namely the control vector $x$, the observation vector $y^{\mathrm{o}}$, the background-error covariance matrix $\mathbf{B}$ and the observation-error covariance matrix $\mathbf{R}$.





Each part of $\mathbf{B}$ corresponding to an optimized flux category is initialized based on the parameters defined in the configuration file. Corresponding eigenvectors and eigenvalues are computed and stored for future usage. Every time the full $\mathbf{B}$ matrix must
be accessed, Kronecker products are used to compute it (see BA21 for further details).

Each member of the ensemble of control vectors must be drawn from a multivariate Gaussian distribution $\mathcal{N}(\boldsymbol{x}^{\mathrm{b}}, \mathbf{B})$. We use the following result to generate this ensemble: if $\boldsymbol{z}$ is an $n$-dimensional vector that follows a multivariate Gaussian distribution $\mathcal{N}(\mathbf{0}, \mathbf{I_n})$, then $\mathbf{C}\boldsymbol{z} + \boldsymbol{\mu}$ follows the distribution $\mathcal{N}(\boldsymbol{\mu}, \mathbf{C}\mathbf{C}^{\mathrm{T}})$, where $\boldsymbol{\mu}$ is a $n$-dimensional vector and $\mathbf{C}$ is a matrix of dimension $n \times n$.

We describe here two simple methods that can be employed to generate $\mathbf{C}$ such that $\mathbf{B} = \mathbf{C}\mathbf{C}^{\mathrm{T}}$. The first method is the Cholesky decomposition, which decomposes $\mathbf{B}$ as $\mathbf{B} = \mathbf{L}\mathbf{L}^{\mathrm{T}}$ where $\mathbf{L}$ is a lower triangular matrix with positive diagonal elements. The second method is a specific application of the so-called Singular Value Decomposition (SVD) method. In our case, it can simply be called eigendecomposition as $\mathbf{B}$ is a square real matrix. This method decomposes $\mathbf{B}$ as $\mathbf{B} = \mathbf{Q}\boldsymbol{\Lambda}\mathbf{Q}^{\mathrm{T}}$ where $\mathbf{Q}$ is an orthogonal matrix whose columns are the orthonormal eigenvectors of $\mathbf{B}$, and $\boldsymbol{\Lambda}$ is a diagonal matrix whose
entries are the eigenvalues of $\mathbf{B}$. As $\mathbf{Q}^{-1} = \mathbf{Q}^{\mathrm{T}}$, we have $\mathbf{C} = \mathbf{Q}\boldsymbol{\Lambda}^{\frac{1}{2}}\mathbf{Q}^{\mathrm{T}}$.

In CIF, we use the second method since the eigenvalues and eigenvectors are already computed and easily accessible. Therefore, we first generate an ensemble of random vectors $\boldsymbol{z_i}$ that each follow a multivariate Gaussian distribution $\mathcal{N}(\mathbf{0}, \mathbf{I_n})$ and then apply the formula $\mathbf{Q}\boldsymbol{\Lambda}^{\frac{1}{2}}\mathbf{Q}^{\mathrm{T}}\boldsymbol{z_i} + \boldsymbol{x}^{\mathrm{b}}$ for each vector using Kronecker products.

The size of the ensemble $N$ is defined by the user in the configuration file. However, the total number of samples that the
CTM needs to transport is $N + 3$ because three "system-bound" samples are inserted at the beginning of the ensemble:

1. The first additional sample is filled with ones only. During the pre-processing of inputs, the CIF routines convert the scaling factors to physical values. This conversion is necessary to ensure that complex operations (e.g., isotope operations on fluxes) can be performed accurately. Subsequently, CIF erases the variables containing the scaling factors to limit memory allocation because most CTMs only need the ensemble of physical fluxes. However, certain models (e.g., ICON-
ART) currently require inputting scaling factors rather than physical fluxes. Therefore, the prior fluxes should always remain accessible to recreate the scaling factors, which is not CIF's default behavior. This is an easy, albeit performant, fix that might be improved in the future.

2. The second additional sample contains the prior values of the scaling factors, which are not necessarily ones.

3. The third additional sample contains the ensemble mean, i.e., the optimized scaling factors. This sample is updated after
each optimization. For the cycle being optimized, it is equal to the background control vector before the optimization and equal to the posterior control vector after the optimization. Note that, before starting the inversion, the second and third additional samples are equal.

We also added multiple optional settings that might be useful in some cases:

–   **Random seed**. Using the same random seed for two different inversions, all the other parameters being equal, will always
generate the same random vectors. If no random seed is selected, a different seed is adopted each time.



- **Adjustment of the mean and variance**. Due to sampling errors, the means and variances of the ensemble may not necessarily align with the means and variances of the corresponding distribution. To rectify this discrepancy, users have the option to enable a setting that adjusts the means and variances, respectively using an offset and a scaling, after the step that generates the random samples.

- **Setting equal prior deviations for all windows**. This technique involves generating the same deviations for all windows at the beginning of the inversion. Consequently, the scaling factors are fully correlated in time. To reproduce the same behavior, users can also choose to utilize a core feature of CIF and prescribe maximal temporal error correlations between different windows directly in the $\mathbf{B}$ matrix and generate the ensemble based on this matrix.

### 3.1.2 Run

The inversion process, as depicted in Fig. 1, involves several steps. We present them here using the example of settings introduced in Sect. 3.1.1.

1. A prior forward simulation of 20 days (10D window length $\times$ 2 nlag) is run with the selected model over the initial cycle (first and second window). A simulation transports one tracer per member of the ensemble plus the three system-bound tracers. Each CTM integrated into CIF possesses its own unique approach to handling these tracers. Notably, users can choose, using a parameter in the configuration file, to transport the full ensemble of tracers within the same simulation or to split this ensemble into multiple simulations if the model cannot accommodate a large number of tracers simultaneously. Simulated values sampled at the locations and times of assimilated observations are provided for each tracer at the end of the simulation.

2. Scaling factors corresponding to the first cycle (first and second windows) are optimized using the outputs of the prior forward simulation and the batch or serial algorithm presented in Sect. 2.3.

3. A posterior forward simulation is run over the first window using the optimized fluxes. This so-called advance step integrates the fluxes of this window to the background concentrations, serving as the starting point for the next cycle.

4. The process moves to the next cycle (second and third window), running again a forward simulation of 20 days with the optimized scaling factors obtained in step 2. All the samples in this simulation are initialized using the final concentrations obtained with the ensemble-mean tracer of the posterior forward simulation performed in step 3.

5. Scaling factors corresponding to the second cycle are optimized. Note that the first window of this cycle (i.e., the second window of the full period) has already been optimized using the observations of the previous cycle in step 2. It is now optimized further using the observations of the third window of the full period.

6. A posterior forward simulation is run over the second window with the optimized scaling factors. This simulation starts from the final concentrations of the posterior forward simulation performed in step 3.






**Figure 1.** Example of the optimization process in CIF with two lags. The full assimilation period is split into M windows and L cycles. In this example, each cycle consists of two windows. The process starts at the lower-left corner with a prior simulation (red box) spanning the first two windows. After the assimilation of observations (red stripes), the posterior simulation (green box) is run until the starting point of the second cycle. The final concentrations of the ensemble-mean tracer obtained with the posterior simulation are used to initialize the next prior simulation (purple arrow). The grey area in the center of the figure and the green and red boxes represent all the windows and cycles that are run between the cycle 3 and the cycle L-2.





The iterative process continues until the last cycle is completed. Each window is simulated nlag+1 times. A larger number of lags may enhance the accuracy if emissions in the present window do not only affect the observations in the present but also in subsequent windows, but it also increases the computation time. It is also important to highlight that the last window is optimized using observations of only one window, regardless of nlag.

### 3.1.3 Localization

Here, we describe how the localization works in CIF. For each window, two distance matrices $\mathbf{L_1}$ and $\mathbf{L_2}$ of dimensions $n \times p$ and $p \times p$ are calculated and applied to the model-observation covariance matrix $\mathbf{X'Y'}^\mathrm{T}$ and the observation-observation covariance matrix $\mathbf{Y'Y'}^\mathrm{T}$, respectively, as described in Sect. 2.3.4. Each element of the first matrix stores the great-circle distance (haversine formula) between the center of the cell or region represented by this element, and the observation's location. Each element of the second matrix stores the great-circle distance between each pair of observation locations. The localization matrices are then calculated using the decay function and length defined by the user in the configuration file. The same decay length is used for both matrices. Four localization functions commonly employed in the ensemble inversion community (Steiner et al., 2024; Peng et al., 2015; Peters et al., 2005; Whitaker and Hamill, 2002) are available in CIF: the Gaussian function, the exponential function, the Heaviside function and the function given by Eq. (4.10) in Gaspari and Cohn (1999), hereafter referred to as the GC99 function. Analytical definitions for these functions are provided in Appendix C.

For the serial EnSRF method, the first localization matrix $\mathbf{L_1}$ is applied to the gain matrix when updating the mean control vector ($\overline{\boldsymbol{x}}$) and the deviations ($\mathbf{X'}$) (see Eq. 27 and Eq. 28). The second matrix $\mathbf{L_2}$ is not applied at this step because $\mathbf{Y'Y'}^\mathrm{T}$ is a scalar. However, it is applied when updating the projection of the mean ($\mathbf{H}\overline{\boldsymbol{x}}$) and deviations ($\mathbf{Y'}$) in the observation space (see Eq. 29 and Eq. 30) to keep consistency between both $\mathbf{X'}$ and $\mathbf{Y'}$ updates. Although we believe this second step is important, it is not described in other EnSRF papers (e.g. CTDAS). Consequently, if both the first and second steps are performed, we call it full localization, as opposed to a partial localization where only the first step is conducted. One of our experiments in Sect. 4 investigates the difference between full and partial localizations.

### 3.1.4 Forecast operator

As described in Section 2.1, the forecast operator is considered either nonexistent or simple in ensemble carbon flux inversions. Initially, Peters et al. (2005) chose to utilize the identity operator when laying the foundation for CTDAS, thereby assuming a maximal correlation between the prior estimate of the control vector for a specific window ($\boldsymbol{x}_w^\mathrm{b}$) and the posterior estimate for the preceding window ($\boldsymbol{x}_{w-1}^\mathrm{a}$), where $w$ denotes the window index. However, in subsequent papers employing the EnSRF algorithm in CTDAS (Steiner et al., 2024; van der Laan-Luijkx et al., 2017; Kim et al., 2014b), the forecast operator was adjusted to a simple weighting function between the posterior estimates of the preceding windows and the original prior estimate of the current window,

$$\boldsymbol{x}_w^{\mathrm{b, updated}} = \sum_{i=1}^{w-1} \lambda_{w-i} \boldsymbol{x}_{w-i}^\mathrm{a} + (1 - \sum_{i=1}^{w-1} \lambda_{w-i}) \boldsymbol{x}_w^\mathrm{b} \tag{33}$$





$\lambda_{w-i}$ are propagation factors ranging between 0 and 1, whose sum is smaller than or equal to 1. The windows in the first cycle are not modified, hence $w \geq$ nlag. This formula is empirical and relies on the assumption that the optimized scaling factors should not vary much from one window to another when the window is reasonably short (e.g., less than a month). Therefore, the information used to update the flux in a window should be partially propagated to the next window. It also mitigates the likelihood of significant discontinuities between fluxes in different windows, especially if the assimilated data is sparser in one window compared to the next one. Also, if the sum of the propagation factors is chosen to be smaller than 1 and the amount of assimilated data drastically drops, then the optimized fluxes will slowly return to prior estimates. In Steiner et al. (2024), a single propagation factor $\lambda_{w-1}$ is used and set to $\frac{2}{3}$. In Kim et al. (2014b) and van der Laan-Luijkx et al. (2017), two propagation factors $\lambda_{w-1}$ and $\lambda_{w-2}$ are used and both set to $\frac{1}{3}$.

This formula has been implemented in CIF-EnSRF. In practice, whenever a new window is about to be optimized for the first time, the associated ensemble mean is updated using Eq. (33) and the samples are shifted based on the difference between the previous and updated ensemble means. Note that the deviations are not modified, hence the prior uncertainties remain identical.

## 3.2 Advantages of the new EnSRF mode

The new EnSRF mode in CIF introduces a wealth of practical features for the inversion community. This value arises not only from recent developments but also from the synergy between the established general features of CIF and the enhanced EnSRF method.

### 3.2.1 Comparison to previous version

Significant enhancements have been made since the original implementation presented in BA21. The initial version featured only a basic structure of the EnSRF method, without the batch optimization method and localization. It also lacked the capability for new cycles to properly restart from previous posterior simulations, preventing a reasonable division of the full assimilation period into multiple windows. Additionally, the pre- and post-processing routines could not handle a large number of samples in a reasonable amount of time and the assimilation process was not optimized computationally, drastically impacting the overall performance of the EnSRF.

To address these limitations, we implemented several key features, including the batch optimization method and localization, bringing the EnSRF method to the level of existing ensemble frameworks. Additionally, we significantly improved the speed of the pre- and post-processing routines within CIF, removing constraints on ensemble size. For each CTM, respective modeling communities can further enhance overall speed by refining routines dedicated to input writing and output processing, e.g. using parallelization. This optimization effort has been done with ICON-ART for this work, and Table D1 provides a breakdown of the time and CPU hours required by both CIF and the ICON-ART model to run the experiments presented in this work.

Lastly, a metrics class has been introduced for EnSRF. This object calculates and stores different types of metrics that are commonly computed in the inversion community and have proven useful in assessing the quality of results. Section 3.3 provides a description of these metrics.





### 3.2.2   Important CIF features

In addition to the new EnSRF features, CIF itself provides a handful of useful core features that were first introduced in BA21 and that work conveniently with the EnSRF mode:

– If the prescribed data is not defined on the same (horizontal or vertical) grid as the selected CTM, then CIF automatically performs the interpolation operations. It can also handle unstructured grids such as the ICON icosahedral grid.

– Multiple categories of emissions for the same species can easily be prescribed and optimized independently.

– The $\mathbf{B}$ matrix is automatically computed based on the configuration file (e.g., flux categories to include, spatial or temporal correlations to calculate, regions to optimize)

– After a potential crash, inversions can resume from any point without any loss of data or time. The only exception is when the CTM fails during one of the forward simulations and is unable to restart directly from the problematic point.

– Any element of the inversion (e.g., prior and posterior fluxes, ensemble of scaling factors, simulated values), for each window and each cycle, is easily accessible.

– Changing the simulated species (e.g., switching from $CH_4$ to $CO_2$) is straightforward, as the variable names and the species attributes are not hard-coded. It only requires a modification of the prescribed data (e.g., surface fluxes, observations, or background concentrations) to ensure consistent results.

– Inversion routines are not model-specific, hence two inversions conducted with two different models undergo identical optimization operations. This core feature helps eliminate many potential discrepancies between elements of an inversion workflow (e.g., pre-processing of prescribed data, CTM run or optimization algorithm). The CIF-EnSRF method has been tested recently with ICON-ART, CHIMERE, and WRF-Chem. Preliminary results from the inversions performed with the three different CTMs appear to be very comparable and, therefore, promising (Berchet et al., 2023).

– CIF can automatically execute complex operations involving different optimized elements, if requested. For example, isotope operations between $\delta^{13}C(CO_2)$ source signatures and $CO_2$ can be performed in order to simulate $^{12}CO_2$ and $^{13}CO_2$, while optimizing $CO_2$ at the end of the simulation.

### 3.3   Metrics

To quantify the success of an inversion, we use different metrics. Most of them are automatically computed by CIF during the inversion. It is important to note that some descriptions are not exhaustive, and for a more comprehensive understanding, references are provided for further exploration.



### 3.3.1 Mean error reduction (MER)

The error reduction (ER) quantifies the agreement between the optimized fluxes and the true fluxes. It is the only metric that is not automatically computed by CIF because it depends on the true scaling factors. It is defined by,

$$\mathrm{ER}(k,t) = 1 - \frac{e^{\mathrm{a}}(k,t)}{e^{\mathrm{b}}(k,t)} \tag{34}$$

$$= 1 - \frac{|x^{\mathrm{a}}(k,t) \cdot F(k,t) - x^{\mathrm{t}}(k,t) \cdot F(k,t)|}{|x^{\mathrm{b}}(k,t) \cdot F(k,t) - x^{\mathrm{t}}(k,t) \cdot F(k,t)|} \tag{35}$$

$x^{\mathrm{b}}(.)$, $x^{\mathrm{a}}(.)$ and $x^{\mathrm{t}}(.)$ are the prior, posterior and true control data (i.e. scaling factors) included in the corresponding vectors $\boldsymbol{x}^{\mathrm{b}}$, $\boldsymbol{x}^{\mathrm{a}}$ and $\boldsymbol{x}^{\mathrm{t}}$, respectively. In this work, $F(.)$ is the respiration flux. $e^{\mathrm{b}}(.)$ and $e^{\mathrm{a}}(.)$ are the prior and posterior absolute flux errors, respectively. $k$ and $t$ represent the cells of the model's horizontal grid and the time dimension, respectively. This formula gives a quantity that is time-dependent and spatially-distributed. We further define the mean error reduction (MER) using an area-weighted spatial average of the flux errors,

$$\mathrm{MER}(t) = 1 - \frac{\overline{e^{\mathrm{a}}(k,t)}}{e^{\mathrm{b}}(k,t)} \tag{36}$$

$$= 1 - \frac{\sum_{k \in \mathcal{S}} a(k) \cdot e^{\mathrm{b}}(k,t)}{\sum_{k \in \mathcal{S}} a(k) \cdot e^{\mathrm{b}}(k,t)} \tag{37}$$

Here, $\mathcal{S}$ represents the CTM's spatial domain or a subpart of this domain (e.g., a country) and $a(.)$ denote the cell's area. A positive MER indicates that the optimized fluxes better agree with the truth than the prior data, whereas a negative MER shows the opposite.

### 3.3.2 Root-mean square deviation (RMSD)

The Root-Mean Square Deviation (RMSD) is commonly employed to quantify the agreement between the observed and simulated atmospheric mole fractions. It is defined by,

$$\mathrm{RMSD} = \sqrt{\frac{1}{p} \sum_{i=1}^{p} (\overline{y}_i - y_i^o)^2} \tag{38}$$

$p$ represents the number of observations, while $\overline{y}_i$ and $y_i^o$ denote the (prior or posterior) simulated and observed value associated to the $i$-th atmospheric observation, respectively. The RMSD can also be computed on a subset of observations, such as specific stations or windows. CIF automatically computes this metric for the full assimilation period and the full set of observations, but also for all assimilated stations, across all the cycles and windows, prior and posterior to the inversion. It should be noted that a lower posterior RMSD does not necessarily mean better performance, since a close agreement with observations can easily be obtained by over-fitting. It is therefore important to combine this metric with others, such as those described below.



### 3.3.3 Cost function reduction (CFR)

The optimal solution derived by the EnSRF minimizes the cost function defined in Eq. (5). To quantify this, we define the cost function reduction (CFR),

$$\text{CFR} = 1 - \frac{J(\boldsymbol{x}^{\text{a}})}{J(\boldsymbol{x}^{\text{b}})} \tag{39}$$

### 3.3.4 Mean uncertainty reduction (MUR)

The EnSRF provides an easy way to calculate the posterior uncertainties using the posterior deviations $\mathbf{X}'^{\text{a}}$, (see Eq. 18). For each cell, we define the uncertainty reduction (UR) as the reduction of the ratio of posterior to prior uncertainties,

$$\text{UR}(x) = 1 - \frac{\sigma^{\text{a}}(k)}{\sigma^{\text{b}}(k)} \tag{40}$$

$\sigma^{\text{b}}(.)$ and $\sigma^{\text{a}}(.)$ denote the prior and posterior standard deviation associated to the cell $k$. We further define the mean uncertainty reduction (MUR) as the average of UR over a domain (e.g. the full domain or a specific country),

$$\text{MUR} = \overline{\text{UR}(k)} \tag{41}$$

Note that it is not the posterior uncertainty of the average but the average of the posterior uncertainty.

### 3.3.5 Reduced chi-squared statistic ($\chi_r^2$)

If the error covariances are properly specified and accurately reflect the true errors in the control variables and the observations, it can be demonstrated that $J(\boldsymbol{x}^{\text{a}})$ has an expected value of $\frac{p}{2}$ (Desroziers and Ivanov, 2001; Talagrand, 1999; Tarantola, 1987). Additionally, if errors are normally distributed, then $J(\boldsymbol{x}^{\text{a}})$ follows a $\chi^2$ distribution with $p$ degrees of freedom and has a standard deviation equal to $\sqrt{\frac{p}{2}}$ (Desroziers and Ivanov, 2001; Talagrand, 1999; Tarantola, 1987). Intuitively, the number of degrees of freedom is $p = n + p - n$ because the number of data points is $n + p$ (prior estimates and observations) and the number of fitted parameters is $n$.

We define the reduced chi-squared statistic $\chi_r^2$,

$$\chi_r^2(\boldsymbol{x}) = \frac{2}{p} J(\boldsymbol{x}) \tag{42}$$

Assuming the previously mentioned assumptions hold, the statistical mean of $\chi_r^2$ over a large number of similar experiments with different perturbations should be equal to 1 and its spread (standard deviation) should be equal to $\sqrt{\frac{2}{p}}$. Consequently, a single experiment should have a $\chi_r^2$ close to 1 when the number of observations is large ($p > 100$). Testing that the $\chi_r^2$ is close to 1 after the inversion therefore provides a simple and low-cost diagnosis for ensuring that the error covariance matrices are properly specified and the ensemble properly approximates the background-error matrix.

### 3.3.6 Degrees of freedom of the ensemble (DOFE)

The DOF (degrees of freedom) of a system refers to the number of independent components within it. In other terms, it represents the number of elements that need to be estimated to obtain a comprehensive understanding of the system. Here, we



employ the formula derived by Fraedrich et al. (1995) and Bretherton et al. (1999), and subsequently employed by Peters et al. (2005) to obtain a statistical estimate of the DOF using the corresponding covariance matrix,

$$\text{DOF} = \frac{(\sum\limits_{i=1}^{n} \lambda_i)^2}{\sum\limits_{i=1}^{n} \lambda_i^2} \tag{43}$$

$\lambda_i$ represents the $i$-th eigenvalue of the covariance matrix defining the system. In our inversion problem, the system of unknown variables is represented by the $\mathbf{B}$ matrix, hence the DOF is obtained by applying this formula to its eigenvalues. The DOF in

the finite ensemble (i.e., obtained by applying the formula to the $\mathbf{B_N}$ matrix) is necessarily smaller than the DOF in our inversion problem (i.e., obtained by applying the formula to the $\mathbf{B_N}$ matrix). Hereinafter, the metric representing the DOF in the ensemble is denoted by DOFE, whereas the DOF in the inversion problem (i.e., the optimal value of the DOFE) is denoted by DOFE$_{\text{opt}}$. For a specific cycle, the closer the DOFE is to the DOFE$_{\text{opt}}$, the closer the EnSRF solution is to the optimal KF solution. Furthermore, one cycle may include multiple windows, hence if the scaling factors representing the different windows

are not correlated in time with each other, the DOF for the cycle is equal to the DOF for a single window multiplied by the number of lags. Conversely, if the scaling factors are fully correlated in time, the DOF for the cycle should be equal to the DOF for a single window.

### 3.3.7 Degrees of freedom for signal (DOFS)

The degrees of freedom for signal (DOFS) quantifies the amount of independent information that can be extracted from the

520 observations to constrain the variables being optimized (Rodgers, 2000). Consequently, higher DOFS leads to more robust estimates. In a general inversion framework, the DOFS is necessarily smaller than $\min(n, p)$. Additionally, with ensemble methods, it cannot exceed the ensemble size without using localization (Hotta and Ota, 2021).

It can be shown that the DOFS is equal to the trace of the so-called *averaging kernel matrix* $\mathbf{A}$ (Brasseur and Jacob, 2017; Rodgers, 2000), which is defined by,

$$\mathbf{A} = \frac{\partial \boldsymbol{x}^{\text{a}}}{\partial \boldsymbol{x}^{\text{t}}} = \frac{\partial \boldsymbol{x}^{\text{a}}}{\partial \boldsymbol{y}^{\text{o}}} \frac{\partial \boldsymbol{y}^{\text{o}}}{\partial \boldsymbol{x}^{\text{t}}} = \mathbf{KH} \tag{44}$$

This matrix represents the sensitivity of the analysis control vector to the true control vector. In an ideal scenario with a perfect observation network, $\mathbf{A}$ would be equal to $\mathbf{I_n}$. While the EnSRF algorithm helps avoid explicit computation of the observation operator $\mathbf{H}$, it also prevents the derivation of $\mathbf{A}$. To circumvent this problem, we also introduce the so-called *influence matrix* $\mathbf{S}^{\text{o}}$ (Cardinali et al., 2004), which is defined by,

$$\mathbf{S}^{\text{o}} = \frac{\partial \boldsymbol{y}^{\text{a}}}{\partial \boldsymbol{y}^{\text{o}}} = \frac{\partial \boldsymbol{y}^{\text{a}}}{\partial \boldsymbol{x}^{\text{a}}} \frac{\partial \boldsymbol{x}^{\text{a}}}{\partial \boldsymbol{y}^{\text{o}}} = \mathbf{K}^{\text{T}}\mathbf{H}^{\text{T}} \tag{45}$$

This matrix represents the sensitivity of the optimized simulated values to the observations. Large diagonal elements (i.e., close to 1) indicate that each observation provided a strong constraint for the corresponding optimized simulated value, compared to the background and the other observations. Using the properties of the trace operator $\text{Tr}(.)$, we have

$$\text{DOFS} = \text{Tr}(\mathbf{A}) = \text{Tr}(\mathbf{KH}) = \text{Tr}(\mathbf{HK}) = \text{Tr}((\mathbf{HK})^{\text{T}}) = \text{Tr}(\mathbf{K}^{\text{T}}\mathbf{H}^{\text{T}}) = \text{Tr}(\mathbf{S}^{\text{o}}) \tag{46}$$



We do not explicity compute $\mathbf{H}$ with the EnSRF, therefore we need another way to compute $\mathbf{S}^{\mathrm{o}}$. Using Eq. (8), Eq. (9) and Eq. (10), we can show that,

$$\mathbf{P}^{\mathrm{a}} = (\mathbf{B}^{-1} + \mathbf{H}^{\mathrm{T}}\mathbf{R}^{-1}\mathbf{H})^{-1} \tag{47}$$

Using this result, we obtain another formulation for $\mathbf{S}^{\mathrm{o}}$,

$$\mathbf{S}^{\mathrm{o}} = \mathbf{K}^{\mathrm{T}}\mathbf{H}^{\mathrm{T}} \tag{48}$$

$$= \mathbf{R}^{-1}\mathbf{H}(\mathbf{B}^{-1} + \mathbf{H}^{\mathrm{T}}\mathbf{R}^{-1}\mathbf{H})^{-1}\mathbf{H}^{\mathrm{T}} \tag{49}$$

$$= \mathbf{R}^{-1}\mathbf{H}\mathbf{P}^{\mathrm{a}}\mathbf{H}^{\mathrm{T}} \tag{50}$$

It shows that the influence matrix is equal to the posterior error covariance matrix mapped onto the observation space and normalized by the observation error covariance matrix. It follows that,

$$\mathrm{DOFS} = \frac{1}{N-1}\mathrm{Tr}(\mathbf{R}^{-1}\mathbf{H}(\mathbf{X'}^{\mathrm{a}})(\mathbf{X'}^{\mathrm{a}})^{\mathrm{T}}\mathbf{H}^{\mathrm{T}}) \tag{51}$$

$$= \frac{1}{N-1}\mathrm{Tr}(\mathbf{R}^{-1}(\mathbf{Y'}^{\mathrm{a}})(\mathbf{Y'}^{\mathrm{a}})^{\mathrm{T}}) \tag{52}$$

This formulation enables an easy computation of the DOFS with the EnSRF (Kim et al., 2014a) at the end of the inversion and after each cycle.

## 4 Demonstration with synthetic experiments

To demonstrate the successful implementation of the new EnSRF method in CIF and the influence of the most important parameters, we present inversion results obtained with different configurations. All examples are synthetic experiments, i.e. inversions assimilating only synthetic observations generated with the CTM and perturbed stochastically. These experiments aim to provide useful guidelines for future inversions utilizing the EnSRF mode of CIF. All experiments in this study cover a two-month period, from 1 June 2018 to 31 July 2018. In addition to these experiments, we also provide in Appendix B a comparison between two inversions with identical setups, one performed with CTDAS and the other with CIF, demonstrating the near-equivalence of the two frameworks.

### 4.1 Configuration of forward simulation

#### 4.1.1 ICON-ART model

The Icosahedral Nonhydrostatic (ICON) Weather and Climate Model (Zängl et al., 2015) is a joint project between the Deutscher Wetterdienst (DWD), the Max-Planck-Institute for Meteorology (MPI-M), the Deutsches Klimarechenzentrum (DKRZ) and the Karlsruhe Institute of Technology (KIT) for developing a unified next-generation global NWP and climate modeling system. The ICON modeling framework became operational in DWD's forecast system in January 2015. Additionally, ICON is being deployed for numerical forecasting for the Swiss meteorological service, MeteoSwiss. ICON has been




released in February 2024 as open source to broaden the community of users and developers. The Aerosols and Reactive Trace gases module (ART), developed and maintained by KIT (Schröter et al., 2018; Rieger et al., 2015), supplements ICON to form

the ICON-ART model, by including emissions, transport, gas phase chemistry, and aerosol dynamics in the troposphere and stratosphere.

ICON-ART is a non-hydrostatic Eulerian CTM. Its horizontal domain is described by an icosahedral grid and can cover either the globe or a limited area, ranging from several degrees to a few kilometers. For this work, a horizontal resolution of 52 km ($\sim 0.7$ °) is adopted for the geographical area covering Europe (15° W – 35° E ; 33° N – 73° N), resulting in a total

number of $c = 5520$ cells. In the vertical, the domain extends from the surface to an altitude of 23 km, with 60 levels described by a height-based terrain-following vertical coordinate.

Meteorological fields are computed online by the ICON model and, in our setup, several prognostic variables (wind speed, specific humidity, density, virtual potential temperature and Exner pressure) are weakly nudged towards the ERA5 reanalysis data (Hersbach et al., 2023, 2017) provided by the ECMWF at a 3-hourly time resolution. This prevents the model from drifting

away from a realistic atmospheric state. The ERA5 data is also employed to initialize the model. For the limited-area mode, boundary conditions can be prescribed at the borders of the domain using external data. Emission fields for any transported species are processed by the Online Emissions Module (OEM, Jähn et al., 2020), included in ART. Output files of instantaneous concentrations are written at hourly resolution and are temporally, vertically and horizontally interpolated offline in order to retrieve simulated equivalents of observations.

### 4.1.2 Input data

**Anthropogenic fluxes**

Anthropogenic $CO_2$ fluxes are based on the spatial distribution of the EDGAR-v4.2 inventory and on national and annual budgets from BP (British Petroleum) statistics. Hourly temporal profiles are derived with the COFFEE approach (Steinbach

et al., 2011, available on the ICOS Carbon Portal). The data is provided at a horizontal resolution of 0.1° × 0.1° and at hourly temporal resolution.

**Biogenic fluxes**

Biogenic $CO_2$ fluxes are derived from ORCHIDEE simulations using two sets of simulations: global simulations from the TRENDY project and higher-resolution simulations from the VERIFY project over Europe. While the latter is used for the region covering (25° W – 45° E; 35 – 73° N), the former allows to extend the domain and cover the full region of interest. More details can be found in Berchet et al. (2023).

**Ocean fluxes**



Ocean $CO_2$ fluxes are derived from a hybrid product that combines the University of Bergen coastal ocean flux estimate with the global ocean estimate from Rödenbeck et al. (2014). This data is available at a horizontal resolution of 0.125° × 0.125° and a daily temporal resolution.


**Background concentrations**

Initial conditions and lateral boundary conditions for $CO_2$ mole fractions are derived from the CAMS global inversion-optimized $CO_2$ concentrations product v20r2 (Chevallier et al., 2010). The data is provided at a horizontal resolution of 3.75°
× 1.9° and at a 3-hourly temporal resolution.

**Observations**

We assimilate synthetic observations matching the observed $CO_2$ atmospheric mixing ratios in Europe compiled in the ver-
sion V8 of the ICOS GlobalView Obspack (ICOS RI et al., 2023). This dataset comprises continuous measurements collected from 58 stations across Europe, including both ICOS and non-ICOS facilities. For the period spanned by our experiments, data from 45 stations are available, as depicted in Fig. 2, and specific information about each observation site is provided in Table D2. The number of synthetic observations to assimilate ($p$) is equal to 12,277.

### 4.1.3  Generation of synthetic data

To create synthetic observations, we first generate a set of scaling factors for each cell of the ICON domain ($c$ = 5520 cells) using the method described in Section 3.1.1. The background-error covariance matrix (of dimension $c \times c$) used for generating the true scaling factors has diagonal elements (variance) equal to 1 (relative variance of 100 %), and the off-diagonal elements (covariance) are calculated based on an exponential decay with a correlation length of 200 km. The resulting scaling factors are shown in Figure 3a. Perturbed fluxes representing the truth are then obtained by applying the scaling factors to the respiration
fluxes, while keeping other fluxes unperturbed. It is important to note that, for the sake of simplicity, the true scaling factors have no time component, and therefore we assume the perturbation to be constant over time. We finally run a forward simulation over the two-month period with the perturbed fluxes.

After this forward simulation, the simulated values matching the assimilated observations are stored. These simulated values are then treated as the new observations to be assimilated in the experiments presented in the next section. However, to mimic
realistic uncertainty in these observations, we perturb them with random values drawn from a Gaussian distribution with a mean of 0 and a standard deviation equal to the observation error calculated for each original observation (see Fig. 2). Note that the resulting observation errors are therefore uncorrelated.

### 4.2  Description of experiments

We categorize the experiments into two groups testing different parameters, LEVEL1 and LEVEL2.





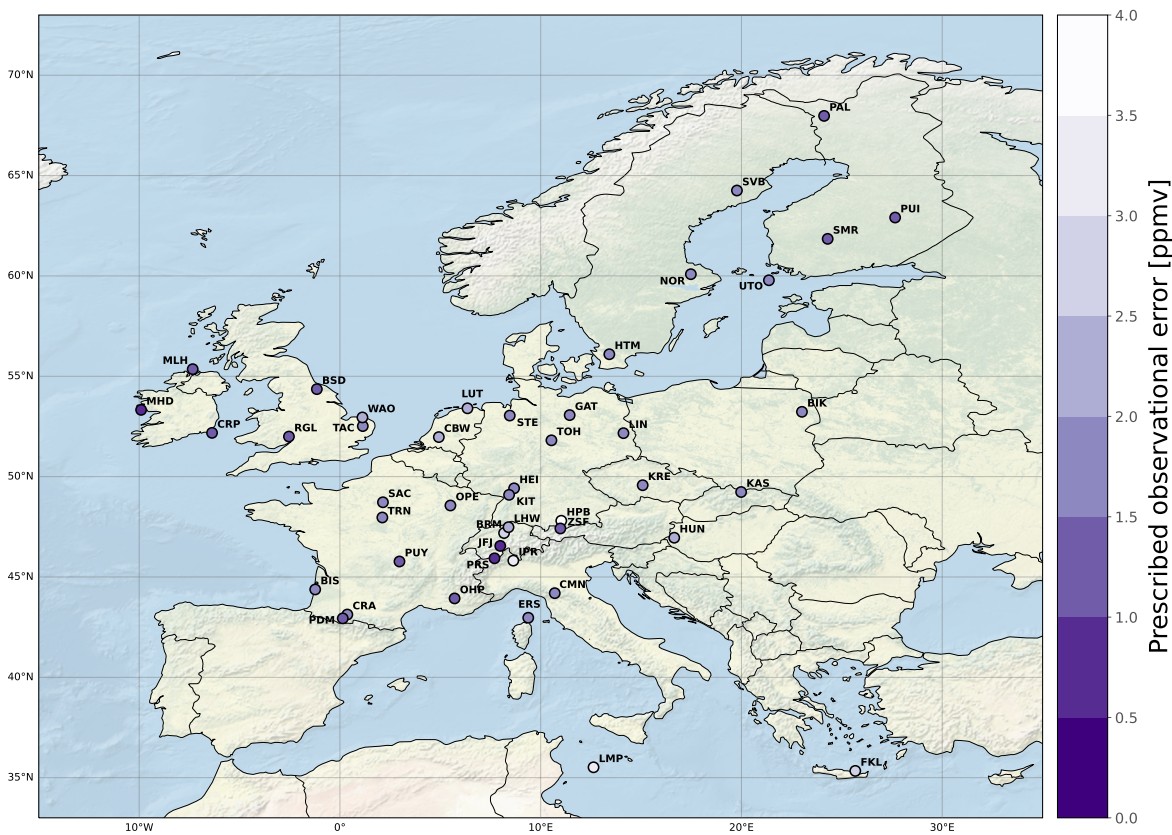

**Figure 2.** Locations of stations assimilated in the synthetic experiments. The purple range shows the observational error prescribed for each station. Background obtained from Natural Earth.

### 4.2.1 LEVEL1 experiments


The LEVEL1 group exclusively assesses the impact of the number of samples and the localization length, recognizing these as critical parameters. We conduct 20 inversions, denoted as N$i$L$j$, where $i$={50, 100, 200, 300} represents the number of samples, and $j$={200,600,1000,1500,none} indicates the localization length in kilometers. For example, N200L600 corresponds to an inversion executed with 200 samples and a localization length of 600 km. N100Lnone corresponds to an inversion executed

with 100 samples but without localization. For each inversion of the LEVEL1 group, the common configuration is provided in Table 1. In particular, we use a window length of 10 days, with two lags, resulting in five cycles of 20 days each.







**Figure 3.** Computation of error reduction for N200L600. a) True scaling factors used to generate the synthetic observations. b) Posterior scaling factors obtained with N200L600 averaged over the full assimilation window. c) Error reduction for each cell. d) MER calculated for each country. This subplot is created by setting each cell's value in a specific country to the MER calculated over the country.



**Table 1.** Description of LEVEL1 experiments. The last column provides the common configuration that all experiments of the LEVEL1 group share.

| Name | Number of samples | Localization length (km) | Other parameters |
|---|---|---|---|
| N$x$L$y$ | $x$={50, 100, 200, 300} | $y$={200, 600, 1000, 1500, none}<br>'none' means no localization is applied. | • Window length = 10 days<br>• Number of lags = 2<br>• Single propagation factor = $\frac{2}{3}$<br>• Localization function = Gaussian function<br>• Horizontal correlation length in the B matrix = 200 km<br>• Random seed for generating samples = 1000<br>• Prior deviations set equal for all windows<br>• Mean and variance of the ensemble not adjusted |

### 4.2.2 LEVEL2 experiments

In the LEVEL2 group, we explore eight additional families of experiments, each denoted by a capital letter, where we test the sensitivity of the EnSRF algorithm to other parameters:

A. We alter the seed used to generate the ensemble to study the impact of randomness on the results.

B. We vary the number of lags.

C. We adjust the propagation factor.

D. We experiment with different localization functions, including exponential, Gaussian, Heaviside, or GC99. All experiments in this family are performed with a localization length of 600 km except for the GC99 case. We observed that the GC99 function is extremely similar to the Gaussian function when the localization length is multiplied by 1.78 (see Figure C1). We therefore use a localization length of 600 × 1.78 = 1068 km to investigate this similarity.

E. We apply either partial localization or full localization.

F. We adjust the mean and variance of the ensemble or not.

G. We set the prior deviations for all windows equal or not.

H. We employ either the serial or the batch EnSRF algorithm.

The LEVEL2 experiments are labeled as EXP_$p$_$v$, where $p$={A, B,..., G, H} is a capital letter representing the tested parameter (i.e. the family), and $v$ represents the value of this parameter. For each family, control experiments are already performed in the LEVEL1 group. Consequently, while the LEVEL2 group comprises 26 experiments, only 16 of them need to be run in addition to the LEVEL1 group. For running inversions with ICON-ART, an ensemble size of 200 is typically employed



to balance computational cost and inversion quality (Steiner et al., 2024). As the best LEVEL1 results with this ensemble size are obtained with a localization length of 600 km, N200L600 is adopted as the control experiment for all families. Some families also feature experiments with a smaller ensemble size when deemed relevant. A summary of all LEVEL2 experiments is presented in Table 2.

**Table 2.** Description of LEVEL2 experiments. Apart from the parameters described in this table, the LEVEL2 experiments all share the same configuration.

| Name | Number of samples | Localization length (km) | Sensitivity parameters |
|---|---|---|---|
| EXP_A_1* | 200 | 600 | Random seed for generating samples = 1000 |
| EXP_A_2* | 200 | 600 | Random seed for generating samples = 2000 |
| EXP_A_3* | 200 | 600 | Random seed for generating samples = 3000 |
| EXP_B_1 | 200 | 600 | Number of lags = 1 |
| EXP_B_2 | 200 | 600 | Number of lags = 2 |
| EXP_B_3 | 200 | 600 | Number of lags = 3 |
| EXP_C_0 | 200 | 600 | Single propagation factor = 0 |
| EXP_C_1 | 200 | 600 | Single propagation factor = $\frac{1}{3}$ |
| EXP_C_2 | 200 | 600 | Single propagation factor = $\frac{2}{3}$ |
| EXP_C_3 | 200 | 600 | Single propagation factor = 1 |
| EXP_D_e | 200 | 600 | Localization function = exponential |
| EXP_D_g | 200 | 1068 | Localization function = GC99 |
| EXP_D_h | 200 | 600 | Localization function = Heaviside |
| EXP_D_n | 200 | 600 | Localization function = Gaussian |
| EXP_E_f | 200 | 600 | Full localization |
| EXP_E_p | 200 | 600 | Partial localization |
| EXP_F_f1* | 50 | 600 | Adjusting the mean and variance of the ensemble = False |
| EXP_F_t1* | 50 | 600 | Adjusting the mean and variance of the ensemble = True |
| EXP_F_f2* | 200 | 600 | Adjusting the mean and variance of the ensemble = False |
| EXP_F_t2* | 200 | 600 | Adjusting the mean and variance of the ensemble = True |
| EXP_G_f | 200 | 600 | Force same prior deviations = False |
| EXP_G_t | 200 | 600 | Force same prior deviations = True |
| EXP_H_b1 | 200 | None | Type of optimization = Batch |
| EXP_H_s1 | 200 | None | Type of optimization = Serial |
| EXP_H_b2 | 200 | 600 | Type of optimization = Batch |
| EXP_H_s2 | 200 | 600 | Type of optimization = Serial |





## 4.3 LEVEL1 results

We explore the impact of ensemble size and localization on our ability to accurately determine the true scaling factors. Since these sensitivities have already been explored extensively in previous EnSRF studies (e.g., Peters et al., 2005), our objective is to validate that our system can produce results consistent with existing literature. Figure 4 illustrates the MER calculated for each country for every LEVEL1 experiment. Across experiments with the same localization length, those with larger ensemble sizes tend to yield scaling factors that align more closely with the true values. For instance, N300Lnone achieves a MER of 13.9

%, whereas N50Lnone exhibits a notably lower value of -21.9 %. Moreover, within experiments sharing the same ensemble size, shorter localization lengths generally yield better results by neglecting long-distance correlations. This localization effect is particularly pronounced in scenarios with smaller ensemble sizes, as evidenced by the improvement from -21.9 % to 23.3 % with 50 samples. However, below a certain threshold, localization can also begin to filter out pertinent information when the number of samples is reasonable. While countries with a dense network of observing sites, such as Western European countries,

benefit from decreasing the correlation length, other countries, such as Balkan or Eastern European countries, show worse results. Overall, with a reasonable number or samples, a localization length of 600 km appears to produce the best results, confirming the results obtained by Peters et al. (2005) with a localization length equal to three times the spatial correlation length prescribed in **B**. Figures 3a and 3b also illustrate an interesting consequence of a sparse network: while the true scaling factor exhibits a patch of values greater than one in the center of Spain, the assimilated observations from this region are not

sufficient to detect it. However, this is not related to the performance of the EnSRF itself.

Figure 4 also shows the posterior $\chi_r^2$ for each experiment, revealing a convergence towards 1 with increased sample size and decreased localization length. Notably, the ratio $J^o/J$ follows the same dependence, namely when the number of samples is low and the localization is weak, only a small part of $J$ is explained by the posterior discrepancies between simulations and observations. It means that there is an excessive distance (defined by the inner product) between the posterior and prior state

vectors, relative to the prescribed uncertainties. An intuitive explanation is that spurious noise in $\mathbf{B_N}$ creates an inconsistency between the characteristics of the optimal solution found by the EnSRF and the expected KF solution that should be obtained with the original **B** matrix.

Figure 5 offers further insights by presenting statistics for individual windows and cycles. Posterior RMSD (Fig. 5b) exhibits substantial consistency across LEVEL1 experiments. Experiments with large ensemble sizes and low localization lengths

slightly outperform others, suggesting that achieving good agreement between posterior simulations and assimilated observations alone does not guarantee high confidence in the results of an EnSRF inversion. Additional diagnostics, such as RMSD calculated with independent observations for real-data inversions or error reductions for synthetic experiments, should be computed. For each window, note that the posterior RMSD closely mirrors the prescribed observation error (Fig. 5b and Fig. 5a, respectively) because the difference between true and estimated fluxes is considerably dampened after the inversion. Addition-

ally, since the true scaling factors remain constant over time and posterior information is partially propagated from one window to the next, the reduction between prior and posterior RMSDs is larger for the initial two windows.





**Figure 4.** MER calculated for each country and for each LEVEL1 experiment. For each experiment, the corresponding subplot is created by setting each cell's value in a specific country to the MER calculated over the country. The corresponding MER calculated for the full domain and the posterior $\chi_r^2$ are displayed in red and blue in the top-left corner, respectively. The value in blue and parenthesis represents the ratio of the $\chi_r^2$ explained by the observational-error part of the cost function ($J^o$), as opposed to the background-error part ($J^b$).





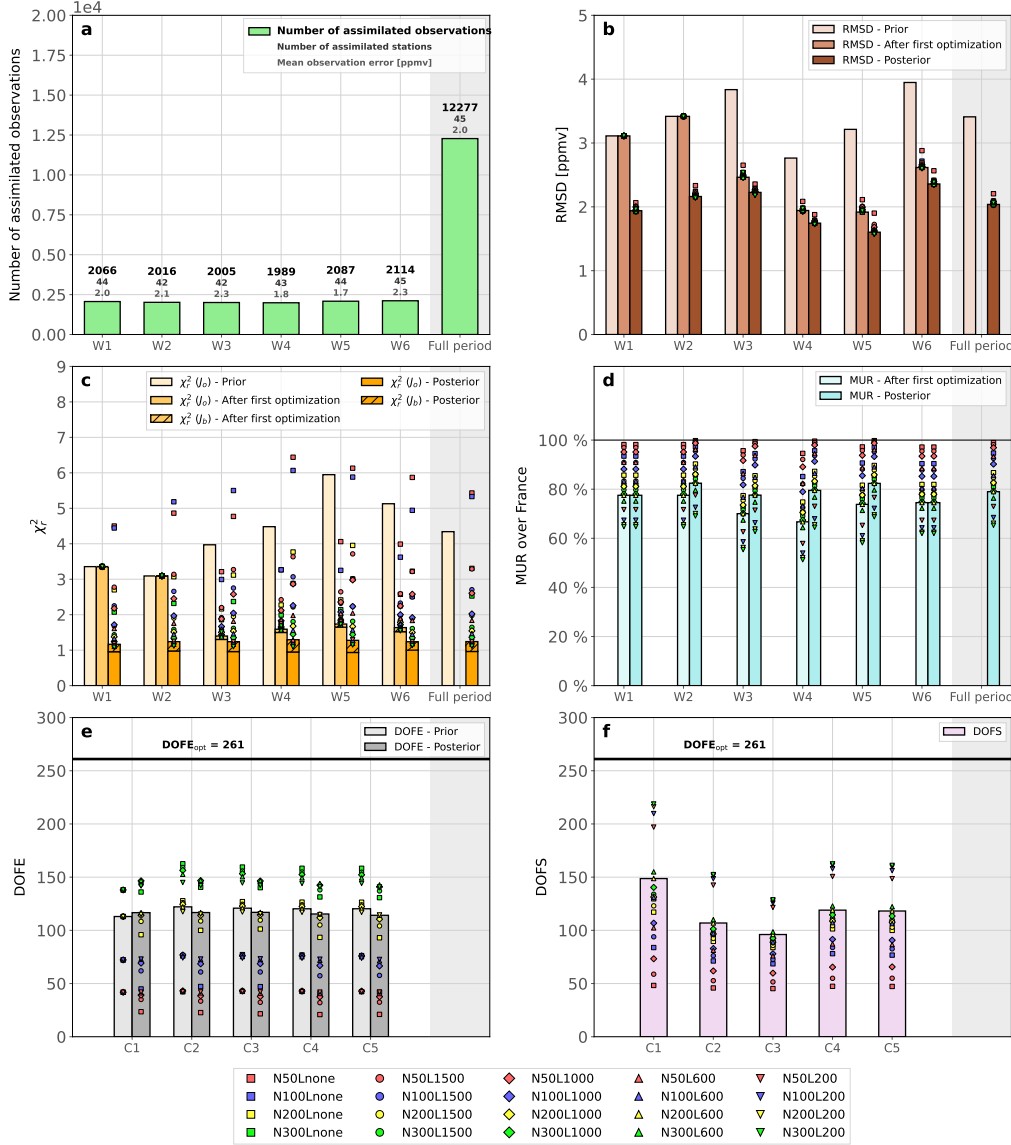

**Figure 5.** Summary of metrics for LEVEL1 experiments. For each panel, each bar represents the N200L600 value of a specific metric for either a window (W), a cycle (C) or the full period. The small markers represent all the LEVEL1 experiments. a) Number of assimilated observations, number of assimilated stations and mean prescribed observation error computed over the assimilated observations in the window. b) Prior, background and posterior RMSD in ppmv. c) Prior, background and posterior $\chi_r^2$. Bars have two components, one for the observation-error part of the cost function and one for the background-error part. d) MUR over France after first and second (posterior) optimizations. e) DOFE computed for each cycle before optimization and after the optimization. f) DOFS computed for each cycle over the assimilated observations after the optimization. Solid black lines in panels e) and f) have been added to show DOFE$_{\text{opt}}$.





Figure 5d shows the MUR averaged over France. We highlight France because it is one of the countries whose metrics are the most affected by the change in the number of samples and localization length. The figure illustrates the tendency of the EnSRF to exhibit increased overconfidence in the derived solution as the number of samples decreases. This underestimation

of the posterior uncertainty was already observed by Peters et al. (2005), Whitaker and Hamill (2002) and Houtekamer and Mitchell (1998). A comprehensive analysis and explanation of this effect is provided by Leeuwen (1999). As the number of samples increases, the posterior uncertainty obtained with the EnSRF tends toward that obtained with the KF. At a constant number of samples, localization helps reduce the bias only for countries with a dense network, while other countries show little or no uncertainty reduction. A spatial illustration is provided in Fig. D1. The consequence is that estimates of EnSRF

posterior uncertainties should be trusted only if the number of samples is reasonably high or if localization is strong enough. Nevertheless, the optimal parameters to employ heavily rely on the inversion problem, and hence sensitivity tests need to be conducted in all cases.

Figures 5e and 5f display the DOFE and DOFS calculated for each cycle. In our LEVEL1 experiments, we find that the DOFE$_{opt}$ for one window equals 261. This number is chiefly controlled not by the number of unknowns (i.e., the spatial

resolution of the model), but by the correlation length, as it is much larger than a model pixel. As we chose to set the deviations of each sample from the mean equal for all windows, the DOFE for the first cycle (i.e., two windows) is equal to the DOFE for a single window. While increasing the DOFE means that we will obtain a solution that is closer to the KF solution for a specific cycle, increasing the DOFS means that more DOF are constrained by observations. The DOFE remains relatively stable throughout the optimization process. As anticipated, this metric increases with the number of samples because $\mathbf{B_N}$

better approximates $\mathbf{B}$. Nevertheless, the DOFE is only around 140 when using 300 samples, significantly lower than the DOFE$_{opt}$. The DOFS also increases with the number of samples, indicating that more DOF are efficiently constrained by the observations. As the DOFS is linked to the posterior uncertainty, a smaller DOFS reflects a larger overconfidence. Additionally, the localization allows to solve the rank-deficiency problem and inflate the DOFS (Hotta and Ota, 2021) but only in the vicinity of observations, as illustrated by the difference between N300Lnone and N300L600. Both DOFE and DOFS offer valuable

insights, yet in cases of low DOFE and with localization, a large DOFS may not necessarily imply a solution closer to reality. This is illustrated by the experiments with a 200 km localization compared to the others. When the localization length is close to the spatial correlation length, non-spurious correlations are also filtered out and the number of apparent DOF in the $\mathbf{B_N}$ matrix surges. Consequently, the number of DOF that seems to be constrained by the observations also increases. For this reason, we recommend 1) using DOFS solely for comparing setups that have identical number of samples and localization

method and 2) always selecting a localization length larger than the spatial correlation.

## 4.4 LEVEL2 results

Table 3 summarizes all the results obtained with the LEVEL2 experiments. Note that we only show the DOFS and DOFE for the first cycle rather than an average or a sum over all cycles because it is easier to interpret. Also, the posterior information is partially propagated from one window to the next, hence the information obtained in the first cycle largely influences the MER

over the full period.



Experiment A investigates the influence of randomness on the results using different seeds. Overall, all the statistics are highly similar. Only the MER and the MUR show small variations of about 1 % and 0.3 % across the experiments, respectively. More tests would be necessary to precisely assess the impact of randomness but this is sufficient to prove that with a reasonable setup, the randomness should not play a significant role.

Experiment B alters the number of lags. As the deviations are set equal for all windows, the DOFE for the first cycle does not change. However, the DOFS for the first cycle increases with the number of lags as more observations are assimilated, and therefore more information is obtained and more DOF are constrained. As the posterior information of the first cycle is partially propagated to the other windows, the MER is larger when selecting more lags. The MUR is also increased because each window is constrained by more observations when we increase the number of lags.

Experiment C explores the propagation of posterior information from one window to the next. Increasing the propagation factors lead to a better MER. As the true scaling factors are constant, the cycles can more effectively build upon the information already collected in the previous cycles if the propagation factor is high. The results might be very different with time-dependent true scaling factors. Note that the MUR calculated over the full domain (not just France) is unchanged because only the mean of the ensemble is propagated forward, not the deviations (i.e. uncertainties). Additionally, a propagation factor of 0 or 1 appears

to slightly reduce the CFR and increase the $\chi_r^2$. As the propagation factor increases, the posterior estimate for a specific window is more likely to diverge further away from the original prior estimate because the system can start the optimization from a point that is already distant, therefore further increasing posterior $J^b(\boldsymbol{x})$ for the window. However, it also reduces $J^o(\boldsymbol{x})$ because the fit to the observations is better. Consequently, there is an optimal value of CFR and $\chi_r^2$ for a propagation factor between 0 and 1 in our case.

Experiment D tests different localization functions. For a localization length of 600 km, the best results are obtained with the exponential function, although the Gaussian function yields similar metrics. The effect of the GC99 function is, as expected, extremely similar to that of the Gaussian function when a factor of 1.76 is applied to the localization length. The Heaviside function gives worse results although it yields a larger DOFS. It is likely because this function is non-smooth and its effect, with a localization length of 600 km, is close to the effect of the Gaussian function with a reduced localization length (e.g.

N200L200). Therefore, it increases the DOFS but the variables are only well constrained in the proximity of the observations and the MER calculated over the full domain is therefore smaller than with the Gaussian and exponential functions. However, these results do not prove that the Heaviside function is generally less performant than the other functions. A larger correlation length could maybe provide similar or better results than the Gaussian and exponential function with a correlation length of 600 km.

Experiment E assesses the difference between full and partial localizations. When the observation-observation covariances are not localized, the results are substantially degraded for our case. Filtering out the long-range observation-observation covariances the same way we do it for the long-range model-observation covariances logically leads to a better consistency between the update of the $\mathbf{Y}'$ (see Eq. 30) and the update of the $\mathbf{X}'$ (see Eq. 28). Our results do not prove that using the full localization is better in all cases. It only confirms that the option to enable or disable the full localization should be easily accessible to users. To the best of our knowledge, other systems do not provide it.






**Table 3.** Description and results for LEVEL2 experiments. A brief description of each experiment and several statistics, such as the MER, MUR, $\chi_r^2$, RMSD, DOFE and DOFS are provided for each experiment of the LEVEL2 group. The MER's value is computed over the full domain for each window and then averaged over all windows. The MUR's value is computed over France and then averaged over all windows. The values of the DOFS and DOFE are given for the first cycle only. The other statistics are computed for the entire assimilation period. * The star symbol indicates that the ensemble generated by CIF for this experiment had to be different from the one generated for the associated control experiment (e.g. N200L600, N200Lnone or N50L600). Therefore, the comparison between experiments is necessarily influenced by randomness.

| Name | Number of samples | Localization length (km) | Other parameters | MER | MUR | RMSD | CFR | $\chi_r^2$ | DOFE | DOFS |
|---|---|---|---|---|---|---|---|---|---|---|
| EXP_A_1* | 200 | 600 | Random seed = 1000 | 27.2 | 79.1 | 2.04 | 71.4 | 1.2 | 113 | 149 |
| EXP_A_2* | 200 | 600 | Random seed = 2000 | 26.2 | 79.3 | 2.04 | 71.3 | 1.2 | 113 | 149 |
| EXP_A_3* | 200 | 600 | Random seed = 3000 | 26.1 | 79.6 | 2.03 | 71.2 | 1.3 | 114 | 149 |
| EXP_B_1 | 200 | 600 | Number of lags = 1 | 22.6 | 71.1 | 2.04 | 73.9 | 1.1 | 113 | 106 |
| EXP_B_2 | 200 | 600 | Number of lags = 2 | 27.2 | 79.1 | 2.04 | 71.4 | 1.2 | 113 | 149 |
| EXP_B_3 | 200 | 600 | Number of lags = 3 | 28.2 | 82.4 | 2.04 | 69.9 | 1.3 | 113 | 181 |
| EXP_C_0 | 200 | 600 | Single propagation factor = 0 | 24.2 | 79.1 | 2.05 | 70.9 | 1.3 | 113 | 149 |
| EXP_C_1 | 200 | 600 | Single propagation factor = $\frac{1}{3}$ | 25.8 | 79.1 | 2.04 | 71.4 | 1.2 | 113 | 149 |
| EXP_C_2 | 200 | 600 | Single propagation factor = $\frac{2}{3}$ | 27.2 | 79.1 | 2.04 | 71.4 | 1.2 | 113 | 149 |
| EXP_C_3 | 200 | 600 | Single propagation factor = 1 | 27.7 | 79.1 | 2.04 | 70.4 | 1.3 | 113 | 149 |
| EXP_D_e | 200 | 600 | Localization function = exponential | 27.6 | 76.6 | 2.04 | 72.8 | 1.2 | 113 | 156 |
| EXP_D_g | 200 | 1068 | Localization function = GC99 | 27.1 | 79.1 | 2.04 | 71.3 | 1.2 | 113 | 149 |
| EXP_D_h | 200 | 600 | Localization function = Heaviside | 22.3 | 75.5 | 2.06 | 68.3 | 1.4 | 113 | 180 |
| EXP_D_n | 200 | 600 | Localization function = Gaussian | 27.2 | 79.1 | 2.04 | 71.4 | 1.2 | 113 | 149 |
| EXP_E_f | 200 | 600 | Full localization | 27.2 | 79.1 | 2.04 | 71.4 | 1.2 | 113 | 149 |
| EXP_E_p | 200 | 600 | Partial localization | 20.7 | 76.7 | 2.44 | 57.3 | 1.9 | 113 | 117 |
| EXP_F_f1* | 50 | 600 | Adjusting the mean/variance = False | 18.3 | 93.5 | 2.07 | 57.6 | 1.8 | 42 | 102 |
| EXP_F_t1* | 50 | 600 | Adjusting the mean/variance = True | 18.1 | 93.4 | 2.07 | 58.0 | 1.8 | 41 | 101 |
| EXP_F_f2* | 200 | 600 | Adjusting the mean/variance = False | 27.2 | 79.1 | 2.04 | 71.4 | 1.2 | 113 | 149 |
| EXP_F_t2* | 200 | 600 | Adjusting the mean/variance = True | 27.3 | 79.1 | 2.04 | 71.4 | 1.2 | 113 | 148 |
| EXP_G_f | 200 | 600 | Force same deviations = False | 23.3 | 78.2 | 2.03 | 72.0 | 1.2 | 144 | 198 |
| EXP_G_t | 200 | 600 | Force same deviations = True | 27.2 | 79.1 | 2.04 | 71.4 | 1.2 | 113 | 149 |
| EXP_H_b1 | 200 | None | Type of optimization = Batch | 6.9 | 86.8 | 2.04 | 23.9 | 3.3 | 113 | 117 |
| EXP_H_s1 | 200 | None | Type of optimization = Serial | 6.9 | 86.8 | 2.04 | 23.9 | 3.3 | 113 | 117 |
| EXP_H_b2 | 200 | 600 | Type of optimization = Batch | 26.7 | 77.5 | 2.04 | 71.9 | 1.2 | 113 | 144 |
| EXP_H_s2 | 200 | 600 | Type of optimization = Serial | 27.2 | 79.1 | 2.04 | 71.4 | 1.2 | 113 | 149 |

Experiment F quantifies the influence of adjusting the ensemble to match the mean and variance of the original distribution. This adjustment has minimal impact on the results, irrespective of whether 50 or 200 samples are used. Furthermore, random-




ness may also play a role because the ensembles are slightly modified by this adjustment. The difference between our results can therefore be considered negligible.

Experiment G investigates the impact of setting prior deviations equal for all windows. In this experiment, the $DOFE_{opt}$ is $261 \times 2 = 522$ in one cycle because the two sets of scaling factors representing each window are not correlated anymore. Although the DOFE is slightly increased from 113 to 144, the ratio of DOFE to $DOFE_{opt}$ becomes smaller. More DOF are constrained by the observations (higher DOFS), but it is not sufficient and the error reduction is larger when the prior deviations are set equal. However, these results may not hold if the true scaling factors are not constant.

Experiment H explores the differences between the batch and serial EnSRF. Without localization and with a diagonal $\mathbf{R}$ matrix, the two algorithms should be mathematically equivalent. They logically produce identical results, indicating that both algorithms are properly implemented. However, the equivalence is broken when localization is applied as expected, although both algorithms provide similar results. CIF therefore allows users to easily leverage the strengths of each algorithm.

### 4.5    Discussion

LEVEL1 experiments have been performed mainly to assess the influence of the ensemble size and localization on the results and LEVEL2 experiments have tested other parameters. The overarching aim of these experiments was to validate that the CIF-EnSRF system produces results consistent with established literature. Here, we have used an oversimplified synthetic setup over Europe to gauge the system's response. The encouraging outcomes we have achieved here represent a crucial, yet not exhaustive, indication of the system's potential efficacy in addressing real-data scenarios. Although certain parameters

may have warranted further experimentation across a broader spectrum of values, we prioritized experiments that we deemed pertinent to highlight the system's capabilities and establish a solid foundation for future research or technical endeavors.

     Future work will nevertheless have to further investigate other aspects of the CIF-EnSRF. First, to better mirror a real-data case, temporal variability needs to be included in the true scaling factors. This temporal variability calls for a better assessment of the potential of prescribing temporal error correlations in the EnSRF. The present work has only begun to delve into this by

examining either maximal temporal correlations or no correlations at all between the windows in the same cycle. There exists a spectrum of other possibilities that warrants further exploration. Also, we only prescribed correlations in $\mathbf{B}$ and $\mathbf{R}$ that were consistent with the parameters used to generate the ensemble, therefore ensuring a perfect characterization of the problem. This is not representative of the reality where variances and covariances are not perfectly known. Including a misrepresentation in the prescribed errors to test the system's response is therefore necessary.

Our current setup focuses solely on optimizing fluxes, overlooking other important components such as initial and boundary conditions. Furthermore, our current setup is limited to the European domain, while the CIF-EnSRF is adaptable to any geographic region, whether regional or global. While this suffices for the scope of our technical demonstration, we recommend conducting tests across various regions and with additional control variables representing other components than fluxes to assess the versatility and performance of this new system.

Finally, the compilation of metrics presented in this work provides a solid foundation for future investigations, facilitating better comparisons between inversions and providing clearer insights into the influence of selected parameters.



## 5 Conclusions

We have presented the new EnSRF mode implemented in CIF. After introducing the theoretical framework of the ensemble inversions and the algorithms we implemented in CIF, we have provided a comprehensive description of the technical im-

plementation in CIF. Finally, we have showcased the enhanced capabilities of CIF-EnSRF using a large number of synthetic experiments, exploring the system's sensitivity to multiple parameters that can be tuned by users.

For inversions conducted over our European domain, employing a spatial correlation length of 200 km in the $\mathbf{B}$ matrix with prior deviations equal across windows yields a DOFE$_{opt}$ of 261. In this case, our synthetic experiments suggest that 200 samples suffice for acceptable results, albeit only when applying localization. Without localization, the spurious correlations

in $\mathbf{B_N}$ have a large influence and lead to poor estimates. Best results are obtained with a localization length of 600km. Note that the DOFE must always be compared to the DOFE$_{opt}$ to confirm whether the number of samples is sufficient. Also, using three lags appears to be slightly more performant than using two lags in our case, although it increases the computational cost. Although a propagation factor of 1 leads to enhanced results in our case, our true scaling factors do not have temporal variability and therefore the results obtained here could be wrong for a real-data case. Consequently, we recommend using a

propagation factor between $\frac{1}{3}$ and $\frac{2}{3}$ and perform sensitivity tests. Finally, using full localization has been beneficial in our case and, as we believe it can be generalized to any case, we recommend using this option.

This work complements previous efforts focused on other inversion methods within CIF. With the successful integration of the EnSRF algorithm, any CTM can now be used to run inversions using CIF, leveraging its capabilities. This enables systematic and rigorous comparisons between different 1) inversion methods and 2) transport models, employing state-of-the-

art techniques. Furthermore, beyond batch and serial EnSRF, there exist other ensemble algorithms utilized in the inversion community. Zupanski et al. (2007) applied a new ensemble method called the maximum likelihood ensemble filter (MLEF) to $CO_2$ regional inversion. Feng et al. (2009) used the revised ETKF from Wang et al. (2004) to derive global $CO_2$ based on satellite data. Chatterjee et al. (2012) developed the Geostatistical Ensemble Square Root Filter (GEnSRF) by modifying the original EnSRF to be consistent with a Geostatistical Inverse Modeling (GIM) formulation of the flux estimation problem

(Michalak et al., 2004). This list is not exhaustive, as other ensemble methods applied to atmospheric inversion exist (e.g., Liu et al., 2022; Peng et al., 2015; Kang et al., 2011, 2012). Our work establishes a groundwork for the integration of these other algorithms, facilitating comparisons and evaluations.

The ever-growing threat imposed by climate change forces the inversion community to produce top-down estimates in near real time. CIF is a powerful tool to reduce the workload associated with running an inversion, while providing robust

estimates and comparisons between them. To validate these inversions, we need to easily access metrics that quantify the success of an inversion, ensure robust comparisons between different inverse modeling teams. The metrics presented in this work are automatically calculated by CIF, for any model, and will be generalized to the other optimization methods in the future. Therefore, we believe this work represents a significant step towards creating an operational system that can address the challenges in GHG emission estimation we are facing today.





As satellite data become more precise and abundant, future work will have to better assess the potential of CIF to take advantage of these rich datasets. Also, at present, the speed of the inversion process can be limited by key routines (e.g., regridding) in CIF if a large number of samples has to be processed with ensemble methods. Consequently, parallelization has to be generalized to these routines. Moreover, CIF greatly benefits from user feedback, and the integration and utilization of new models (ICON-ART, WRF-CHEM, STILT, etc.), will contribute to shaping a system that caters to the needs of members

within the inversion community and facilitate policy-relevant research on greenhouse gas emissions.



## Appendix A: Notations

**Table A1.** Summary of all notations used in this paper.

| Notation | Description | Dimensions |
|---|---|---|
| $n$ | Total number of optimized variables | scalar |
| $m$ | Number of optimized variables in a window | scalar |
| $p$ | Total number of observations | scalar |
| $c$ | Number of cells in the ICON horizontal domain | scalar |
| $N$ | Number of samples (or members) in the ensemble | scalar |
| $\boldsymbol{x}$ | Control (or state or target) vector | $n$ |
| $\boldsymbol{x}^{\mathrm{b}}$ | Prior (or background) control vector | $n$ |
| $\boldsymbol{x}^{\mathrm{a}}$ | Posterior (or analysis) control vector | $n$ |
| $\boldsymbol{y}^{\mathrm{o}}$ | Observation vector | $p$ |
| $\boldsymbol{\epsilon}^{\mathrm{b}}$ | Background error | $n$ |
| $\boldsymbol{\epsilon}^{\mathrm{a}}$ | Analysis error | $n$ |
| $\boldsymbol{\epsilon}^{\mathrm{o}}$ | Observation error | $p$ |
| $\mathbb{E}[.]$ | Expectation operator | any $\mapsto$ any |
| $\mathbf{B} = \mathbb{E}[(\boldsymbol{\epsilon}^{\mathrm{b}})(\boldsymbol{\epsilon}^{\mathrm{b}})^{\mathrm{T}}]$ | Background-error covariance matrix | $n \times n$ |
| $\mathbf{P}^{\mathrm{a}} = \mathbb{E}[(\boldsymbol{\epsilon}^{\mathrm{a}})(\boldsymbol{\epsilon}^{\mathrm{a}})^{\mathrm{T}}]$ | Analysis-error covariance matrix | $n \times n$ |
| $\mathbf{R} = \mathbb{E}[(\boldsymbol{\epsilon}^{\mathrm{o}})(\boldsymbol{\epsilon}^{\mathrm{o}})^{\mathrm{T}}]$ | Observation-error covariance matrix | $p \times p$ |
| $\mathcal{H}(.)$ | Observation operator | $n \mapsto p$ |
| $\mathbf{H}$ | Jacobian matrix of the observation operator | $p \times n$ |
| $\mathbf{K}$ | Kalman gain matrix | $n \times p$ |
| $J(.)$ | Cost function operator | $n \longrightarrow$ scalar |
| $J^{\mathrm{o}}(.)$ | Observation part of the cost function operator | $n \mapsto$ scalar |
| $J^{\mathrm{b}}(.)$ | Background part of the cost function operator | $n \mapsto$ scalar |
| $\mathcal{F}(.)$ | Forecast operator (or propagation operator) | $m \mapsto m$ |
| $\mathcal{N}(\boldsymbol{\mu}, \mathbf{C})$ | Multivariate Gaussian distribution of mean $\boldsymbol{\mu}$ and covariance matrix $\mathbf{C}$ | n/a |
| $\overline{\boldsymbol{x}}$ | Mean of the Gaussian distribution of control vectors | $n$ |
| $\mathbf{X} = (\boldsymbol{x_1}, ..., \boldsymbol{x_N})$ | Sample control vectors | $n \times N$ |
| $\mathbf{X}' = (\boldsymbol{x_1} - \overline{\boldsymbol{x}}, ..., \boldsymbol{x_N} - \overline{\boldsymbol{x}})$ | Deviations of the sample control vectors from the mean control vector. | $n \times N$ |
| $\mathbf{B_N} = \frac{1}{N-1}\mathbf{X}'\mathbf{X}'^{\mathrm{T}}$ | Ensemble approximation of background-error covariance matrix | $N \times N$ |
| $\mathbf{Y}' = \mathbf{H}\mathbf{X}'$ | Projection of $\mathbf{X}'$ in the observation space | $p \times N$ |
| $\boldsymbol{d} = \boldsymbol{y}^{\mathrm{o}} - \mathbf{H}\overline{\boldsymbol{x}}$ | Innovation vector | $p$ |
| $\mathbf{D} = \frac{1}{N-1}\mathbf{Y}'\mathbf{Y}'^{\mathrm{T}} + \mathbf{R}$ | Innovation covariance matrix | $p \times p$ |
| $\mathbf{T} = \mathbf{I_N} - \frac{1}{N-1}\mathbf{Y}'^{\mathrm{T}}\mathbf{V}\mathbf{Y}'$ with $\mathbf{V} = \mathbf{D}^{-\frac{1}{2}}(\mathbf{D}^{\frac{1}{2}} + \mathbf{R}^{\frac{1}{2}})^{-1}$ | Transformation matrix for EnSRF | $N \times N$ |
| $\mathbf{A} = \frac{\partial \boldsymbol{x}^{\mathrm{a}}}{\partial \boldsymbol{x}^{\mathrm{t}}} = \mathbf{K}\mathbf{H}$ | Averaging kernel matrix | $n \times n$ |
| $\mathbf{S}^{\mathrm{o}} = \frac{\partial \boldsymbol{y}^{\mathrm{a}}}{\partial \boldsymbol{y}^{\mathrm{o}}} = \mathbf{K}^{\mathrm{T}}\mathbf{H}^{\mathrm{T}}$ | Influence matrix | $n \times n$ |



## Appendix B: Comparison between CTDAS and CIF

CTDAS is widely used within the EnSRF inversion community. As part of our study, we conducted a comparison between CTDAS and CIF to demonstrate to inverse modelers that the EnSRF algorithm implemented in CIF is equivalent to the one in CTDAS. To provide a robust comparison, we needed to perform two inversions with identical setups, data processing, and assimilation methods (e.g., the same assimilation order). We selected the reference synthetic inversion (case 1) of $CH_4$ fluxes over Europe performed by Steiner et al. (2024) with CTDAS and aimed to replicate it precisely with CIF. As the complete setup is detailed in the associated paper, we only outline the main components of the configuration here:

- The assimilation window spans the period 2018-01-01 to 2018-03-01.

- The ICON horizontal resolution is R3B6 ($\sim 26$ km)

- The ensemble size is 192.

- The number of lags is 2.

- The window length is 10 days.

- The localization is applied with a Gaussian function and a length of 600 km.

- Two emission categories are optimized: anthropogenic and natural.

- The number of optimized variables for each category is equal to 21344, which is the number of horizontal cells in the ICON domain.

- A relative prior uncertainty of 100 % is prescribed for each emission category.

- The synthetic observations are generated using a forward simulation spanning the full assimilation window and applying a random noise of 2 ppbv to the simulated values at the observation locations.

- The diagonal elements of the observation-error matrix are equal to 10 ppbv + 30 % of the prior signal of the $CH_4$ emissions, at the corresponding observation's location, averaged over the entire inversion period. Observation errors are uncorrelated.

The same ensemble (i.e., same ensemble size and same sample values) is employed in both inversions to eliminate the influence of randomness. We endeavored to replicate the post-processing of ICON outputs (i.e., time, horizontal and vertical interpolations) as closely as possible in CIF. However, despite our efforts, minor differences (less than 2 ppbv) sometimes emerged between the $CH_4$ mole fractions simulated by the two systems. This discrepancy is particularly noteworthy because the synthetic observations are generated using the CTDAS version described by Steiner et al. (2024). Consequently, the posterior scaling factors obtained with CIF cannot be as close to the truth as those obtained with CTDAS. Despite these caveats, Figures B1 and B2 show very similar results between CTDAS and CIF for both emission categories and for both the error reduction and the posterior uncertainties.



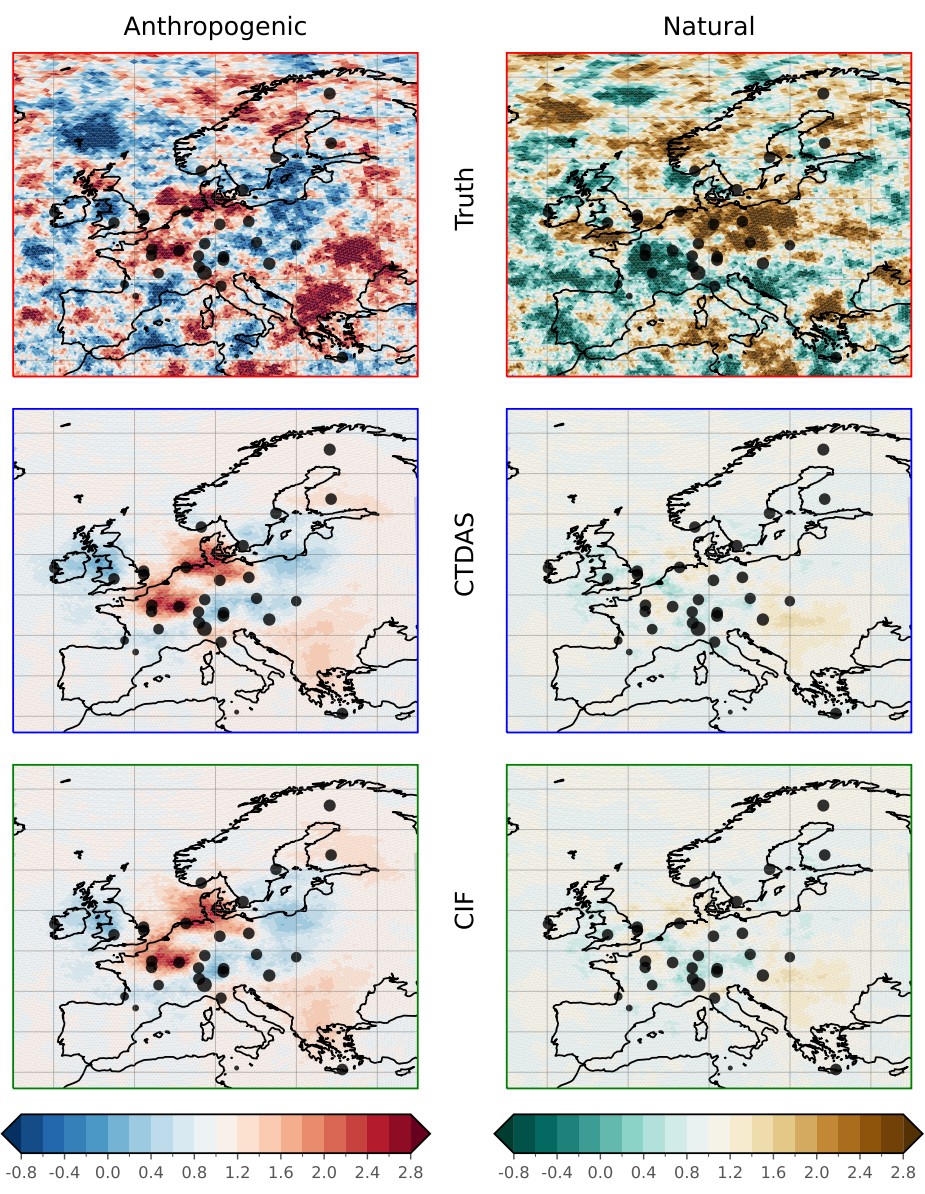

**Figure B1.** True and optimized scaling factors averaged over the full assimilation period for the anthropogenic (left panels) and natural (right panels) emission categories. The upper panels represent the true scaling factors applied to the fluxes to generate the synthetic observations. The center panels represent the optimized scaling factors obtained by Steiner et al. (2024) using CTDAS. The lower panels represent the optimized scaling factors obtained using CIF and the same inversion setup that lead to the CTDAS scaling factors. The dots represent the locations of observation sites. Their size is proportional to the number of assimilated observations provided by the corresponding site.

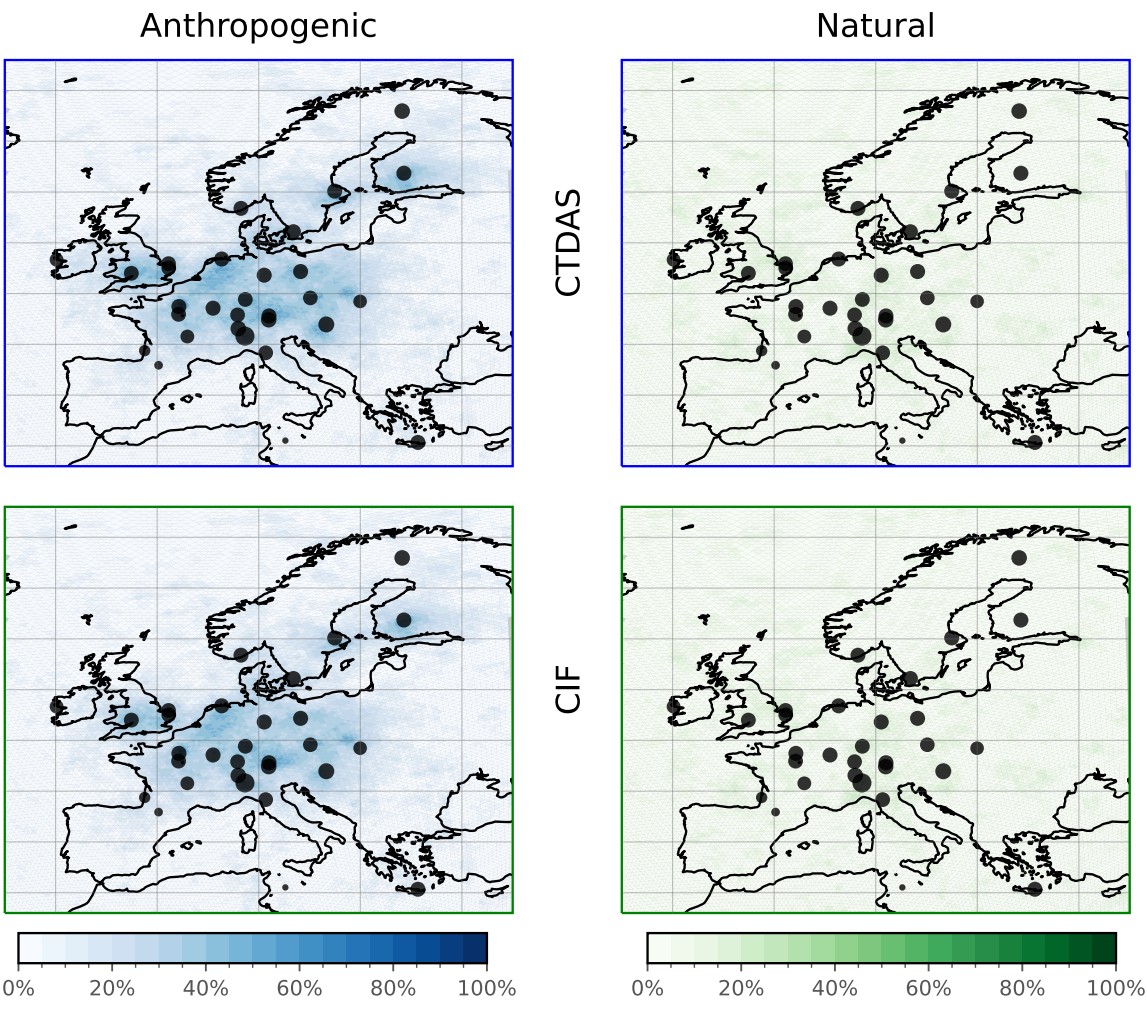

**Figure B2.** Uncertainty reductions averaged over the full assimilation period for the anthropogenic (left panels) and natural (right panels) emission categories. The upper panels represent the uncertainty reduction obtained by Steiner et al. (2024) using CTDAS . The lower panels represent the optimized scaling factors obtained using CIF.



## Appendix C: Localization functions

We introduced four localization functions in Sect. 3.1.3: the Gaussian, exponential, Heaviside and GC99 functions. For each function, an analytical definition and an illustration (Fig. C1) are provided below. If $d$ denotes the great-circle distance between two locations on Earth, $l$ the localization length and $r = d/l$, the localization functions are defined by:

$$L_{\text{Gaussian}}(r) = e^{-\frac{r^2}{2}} \tag{C1}$$

$$L_{\text{exponential}}(r) = e^{-r} \tag{C2}$$

$$L_{\text{Heaviside}}(r) = \begin{cases} 1 & \text{if } r \leq 1 \\ 0 & \text{if } r > 1 \end{cases} \tag{C3}$$

$$L_{\text{GC99}}(r) = \begin{cases} -\frac{1}{4} \cdot r^5 + \frac{1}{2} \cdot r^4 + \frac{5}{8} \cdot r^3 - \frac{5}{3} \cdot r^2 + 1 & \text{if } 0 \leq r \leq 1 \\ \frac{1}{12} \cdot r^5 - \frac{1}{2} \cdot r^4 + \frac{5}{8} \cdot r^3 + \frac{5}{3} \cdot r^2 - 5 \cdot r + 4 - \frac{2}{3} \cdot r^2 - 1 & \text{if } 1 < r \leq 2 \\ 0 & \text{if } 2 < r \end{cases} \tag{C4}$$

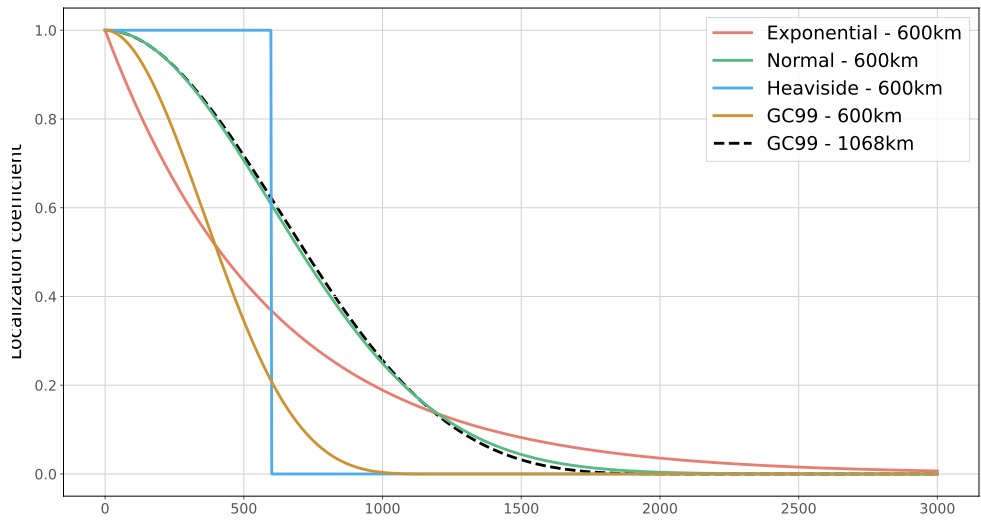

**Figure C1.** Localization coefficient as a function of distance (in km). Four functions with a localization length of 600 km are displayed: exponential (red), Gaussian (green), Heaviside (blue) and GC99 (yellow). The black dashed line represents the function GC99 with a localization length of 600 x 1.78 = 1068 km to highlight that the GC99 function is highly similar to the Gaussian function when a coefficient of 1.78 is applied to the localization length of the Gaussian function.





**Appendix D: Additional figures and tables**

**Figure D1.** Uncertainty reduction calculated for each LEVEL1 experiment. The corresponding MUR calculated for the full domain is displayed in black in the top-left corner.





**Table D1.** CPU hours used by CIF and ICON-ART (resolution R3B5 with 5520 cells) to perform LEVEL1 experiments (two-month period)on the supercomputer Piz Daint at the Swiss National Supercomputing Center (CSCS).

| Ensemble size | CPU hours used by CIF | CPU hours used by ICON-ART |
|---|---|---|
| 50 | 0.6 | 288 |
| 100 | 1.0 | 432 |
| 200 | 1.3 | 936 |
| 300 | 2.5 | 1728 |





**Table D2.** Overview of the surface stations that provided the assimilated observations.

| ID | Name | Country | Latitude | Longitude | Altitude (m.a.s.l.) | Inlet height (m.a.g.l.) |
|---|---|---|---|---|---|---|
| BIK | Bialystok | PL | 53.23 | 23.01 | 183 | 300 |
| BIS | Biscarrosse | FR | 44.38 | -1.23 | 73 | 47 |
| BRM | Beromunster | CH | 47.19 | 8.18 | 797 | 212 |
| BSD | Bilsdale | GB | 54.36 | -1.15 | 382 | 248 |
| CBW | Cabauw | NL | 51.97 | 4.93 | 0 | 207 |
| CMN | Monte Cimone | IT | 44.19 | 10.70 | 2165 | 8 |
| CRA | Centre de Recherches Atmosphériques | FR | 43.13 | 0.37 | 600 | 60 |
| CRP | Carnsore Point | IE | 52.18 | -6.37 | 9 | 14 |
| ERS | Ersa | FR | 42.97 | 9.38 | 533 | 40 |
| FKL | Finokalia | GR | 35.34 | 25.67 | 250 | 15 |
| GAT | Gartow | DE | 53.07 | 11.44 | 70 | 341 |
| HEI | Heidelberg | DE | 49.42 | 8.68 | 113 | 30 |
| HPB | Hohenpeissenberg | DE | 47.80 | 11.02 | 934 | 131 |
| HTM | Hyltemossa | SE | 56.10 | 13.42 | 115 | 150 |
| HUN | Hegyhátsál | HU | 46.96 | 16.65 | 248 | 115 |
| IPR | Ispra | IT | 45.81 | 8.64 | 210 | 100 |
| JFJ | Jungfraujoch | CH | 46.55 | 7.99 | 3572 | 14 |
| KAS | Kasprowy Wierch | PL | 49.23 | 19.98 | 1987 | 7 |
| KIT | Karlsruhe | DE | 49.09 | 8.42 | 110 | 200 |
| KRE | Křešín u Pacova | CZ | 49.57 | 15.08 | 534 | 250 |
| LHW | Laegern-Hochwacht | CH | 47.48 | 8.40 | 840 | 32 |
| LIN | Lindenberg | DE | 52.17 | 14.12 | 73 | 98 |
| LMP | Lampedusa | IT | 35.52 | 12.63 | 45 | 8 |
| LUT | Lutjewad | NL | 53.40 | 6.35 | 1 | 60 |
| MHD | Mace Head | IE | 53.33 | -9.90 | 5 | 24 |
| MLH | Malin Head | IE | 55.36 | -7.33 | 22 | 47 |
| NOR | Norunda | SE | 60.09 | 17.48 | 46 | 100 |
| OHP | Observatoire de Haute Provence | FR | 43.93 | 5.71 | 650 | 100 |
| OPE | Observatoire pérenne de l'environnement | FR | 48.56 | 5.50 | 390 | 120 |
| PAL | Pallas | FI | 67.97 | 24.12 | 565 | 12 |
| PDM | Pic du Midi | FR | 42.94 | 0.14 | 2877 | 28 |
| PRS | Plateau Rosa | IT | 45.93 | 7.70 | 3480 | 10 |
| PUI | Puijo | FI | 62.91 | 27.65 | 232 | 84 |
| PUY | Puy de Dôme | FR | 45.77 | 2.97 | 1465 | 10 |
| RGL | Ridge Hill | GB | 52.00 | -2.54 | 207 | 90 |
| SAC | Saclay | FR | 48.72 | 2.14 | 160 | 100 |
| SMR | Hyytiälä | FI | 61.85 | 24.29 | 181 | 125 |
| STE | Steinkimmen | DE | 53.04 | 8.46 | 29 | 252 |



**Table D2.** Following Table D2.

| ID | Name | Country | Latitude | Longitude | Altitude (m.a.s.l.) | Inlet height (m.a.g.l.) |
|---|---|---|---|---|---|---|
| STE | Steinkimmen | DE | 53.04 | 8.46 | 29 | 252 |
| SVB | Svartberget | SE | 64.26 | 19.78 | 269 | 150 |
| TAC | Tacolneston | GB | 52.52 | 1.14 | 64 | 185 |
| TOH | Torfhaus | DE | 51.81 | 10.54 | 801 | 147 |
| TRN | Trainou | FR | 47.96 | 2.11 | 131 | 180 |
| UTO | Utö - Baltic sea | FI | 59.78 | 21.37 | 8 | 57 |
| WAO | Weybourne | GB | 52.95 | 1.12 | 31 | 10 |
| ZSF | Zugspitze | DE | 47.42 | 10.98 | 2666 | 3 |



*Code and data availability.* The ICON and ART codes are now open source and publicly available for download at https://doi.org/10.35089/WDCC/IconRelease01 (ICON Partnership, 2024). The CIF code featuring the new EnSRF mode can be accessed via the following DOI: https://doi.org/10.5281/zenodo.12742377 (Berchet et al., 2022), while input data (fluxes, background concentrations and observations) are publicly available via the following DOI: https://doi.org/10.5281/zenodo.12609041 (Berchet et al., 2024). Complete and surface ERA5
reanalysis data are publicly available via the Copernicus Climate Change Service at https://doi.org/10.24381/cds.143582cf (Hersbach et al., 2017) and https://doi.org/10.24381/cds.adbb2d47 (Hersbach et al., 2023), respectively.

*Author contributions.* AB and AT initially implemented the foundation of the EnSRF mode in CIF. LC laid the groundwork for coupling CIF and ICON-ART. JT enhanced the CIF-EnSRF mode as described in this paper, completed the coupling between CIF and ICON-ART and implemented the metrics routines. FR tested the EnSRF mode with the WRF model and contributed to the improvement of the code. AB
contributed his technical expertise on CIF and scientific expertise on inversions. MS, FR, DB, and SH contributed their scientific expertise on EnSRF inversions. JT ran the experiments and led the manuscript preparation, with input and contributions from all co-authors.

*Competing interests.* The authors declare they have no competing interests.

*Acknowledgements.* This project has received funding from the European Union's Horizon 2020 programme under grant agreement no. 958927 (CoCO2) and the European Union's Horizon Europe programme under grant agreement no. 101081322 (AVENGERS). The CIF-
ICON-ART inversions were conducted at the Swiss National Supercomputing Centre (CSCS) under grant No. s1152 and were supported by the Center for Climate Systems Modeling (C2SM). The authors thank Vladislav Bastrikov (Science Partners) and the ORCHIDEE project team for providing the input $CO_2$ fluxes.



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
