# Peer review of "Improving the EnSRF in the Community Inversion Framework: a case study with ICON-ART 2024.01"

_EGUsphere, 2024_

## Author Comment (AC1)

**Response to Referees' Comments**

**Joël Thanwerdas[1], Antoine Berchet[2], Lionel Constantin[1], Aki Tsuruta[3], Michael Steiner[1], Friedemann Reum[4], Stephan Henne[1], and Dominik Brunner[1]**

[1]Empa, Swiss Federal Laboratories for Materials Science and Technology, Dübendorf, Switzerland.

[2]Laboratoire des Sciences du Climat et de l'Environnement, CEA-CNRS-UVSQ, IPSL, Gif-sur-Yvette, France.

[3]Finnish Meteorological Institute (FMI), Helsinki, Finland

[4] Deutsches Zentrum für Luft- und Raumfahrt e.V., Institut für Physik der Atmosphäre, Oberpfaffenhofen, Germany

We thank the two referees for their invaluable insights, which have greatly enhanced the quality of the paper. We provide here a comprehensive response to the comments received. Referee#1 (Arne Babenhauserheide)'s comments are in red and Referee#2's comments are in blue. For each comment, an answer is provided in normal text and **the modifications from the new version of the manuscript are provided in bold and small text**.

Attached to this response, we also provide the new version of the manuscript and a track-changes document.

**Specific comments**

**Abstract**

- L 5: "more efficient implementation": Table D1 shows a benchmark of the new code. Is there a benchmark of the old code to support this description?

Unfortunately no, there is no benchmark of the old code. For real-data inversions, a horizontal resolution of 26 km over Europe is commonly used (21,344 cells). With this resolution, ICON-ART could not be used with a reasonable number of samples (i.e. > 50) with the old code due to a bad management of memory.

We slightly changed this sentence in the abstract to make it more accurate.

**In this paper, we present and evaluate a new implementation of the ensemble mode, building upon the initial developments.**

- L 15: what does ICON-ART bring as advantage? Why does this use ICON-ART? It is in the title, so it should be in the abstract.

While the ICON-ART model is highlighted in the manuscript title, it is not mentioned in the abstract or introduction. It might be helpful to include at least a brief mention of ICON-ART in these sections.

As these questions are similar, we answer both of them here. These new developments started when the need to run the CIF with ICON-ART and WRF-CHEM became more pressing in the EU Horizon 2020 project CoCO$_2$.

[Figure]

As ICON-ART and WRF-CHEM are typically used with ensemble methods and adjoints are difficult to code and maintain, the ensemble mode in the CIF needed to be improved.

ICON-ART is a very flexible and scalable non-hydrostatic atmospheric transport model, which can be run from global to local scales and which offers advanced features such as flexible (including two-way) nesting and perfect local mass conservation. For the purpose of this study, however, any other CTM could have been used, such as WRF-CHEM.

Also, GMD is requesting authors to provide the name of the model in the title in a "development and technical paper". It is stated that: "*If the main intention of an article is to make a general (i.e. model independent) statement about the usefulness of a new development, but the usefulness is shown with the help of one specific model, the model name and version number must be stated in the title*". If this request was not clearly formulated, we would have chosen to remove it from the title.

We added some details about ICON-ART in the abstract and the introduction.

Abstract: **Finally, we demonstrate the capabilities of the CIF-EnSRF system using a large number of synthetic experiments over Europe with the flexible and scalable high-performance atmospheric transport model ICON-ART, exploring the system's sensitivity to multiple parameters that can be tuned by users.**

Abstract: **This work complements previous efforts focused on other inversion methods within CIF. While ICON-ART has been used for testing in this work, the integration of these new ensemble algorithms enables any chemical transport model (CTM) to perform inversions, fully leveraging CIF's robust capabilities.**

Introduction: **This method therefore needed improvements, which were initiated when performing $CO_2$ inversions with CIF using different models and inversion setups as part of the Horizon 2020 CoCO2 project (Berchet et al., 2023). The model ICON-ART (Zängl et al., 2015; Rieger et al., 2015), described in Sect. 4.1.1, was one of these models. It is utilized here to showcase the capabilities of the new ensemble method in CIF.**

**Introduction**

- L 39: very nice description! (nothing to change, just a note)

Thanks a lot!

- L 74: please add a short overview of numerical weather prediction (NWP), tracer models, ICON, and ART. The reports on ICON may be a good resource; either to cite relevant reports, or to add a link in a footnote: https://www.dwd.de/DE/leistungen/reports_on_icon/archiv/archiv_reports_on_icon.html. Maybe note that more details are provided in 4.1.1.

Following the previous comment about including ICON-ART in the introduction, we added a short sentence about ICON-ART in this introduction, addressing the previous comment about mentioning the model in this part. As the referee stated, we note that more details are provided in Sect. 4.1.1. Our focus is on our novel developments in CIF, not least because - as mentioned above – our method can be used with many transport models. Therefore, while understanding NWP and tracer models is important for our field, we believe the brief overview about data assimilation methods in the introduction is sufficient for the purpose of our paper. Also,

we feel that the ICON reports the referee has suggested to cite are highly technical and not suitable for a general overview about NWP and tracer models. We prefer mentioning the main references of ICON and ART which also introduce very well the concepts of NWP and tracer models.

Introduction: **This method therefore needed improvements, which were initiated when performing CO2 inversions with CIF using different models and inversion setups as part of the Horizon 2020 CoCO2 project (Berchet et al., 2023). The model ICON-ART (Zängl et al., 2015; Rieger et al., 2015), described in Sect. 4.1.1, was one of these models. It is uti-lized here to showcase the capabilities of the new ensemble method in CIF.**

**Section 2.2**

- L 178: what happens in case of nonlinear effects like OH-destruction of CH4? This is not a problem, but should be noted.

If OH concentrations are not optimized (i.e., included in the control vector) the observation operator H is still linear and the same Kalman Filter (KF) equations can be applied. If OH concentrations are optimized, then the observation operator H is nonlinear. In this case, KF equations cannot be applied the way it is described in the paper. One needs to use an extended Kalman filter as mentioned in line 121, in section 2.1. With this filter, the observation operator needs to be linearized around the background control vector.

We added these considerations in Sect. 2.1, where the extended Kalman Filter is introduced, rather than in Sect 2.2. Additionally, we removed the mention of H being linear in Sect. 2.3.2, as it is implicitly assumed that H is linear when applying these equations.

Sect. 2.1: **For inversion applications, this filter consists simply in linearizing H around the background control vector to be able to apply the KF equations. For example, consider a scenario in which $CH_4$ is transported by the CTM, and its reaction with OH, the primary $CH_4$ sink, is included in the model. If both $CH_4$ emissions and OH concentrations are treated as optimized variables (i.e., included in the control vector), the observation operator becomes nonlinear. In this case, the observation operator must be linearized before applying the KF equations. However, if OH concentrations are not included in the optimization process, the observation operator remains linear, allowing the implementation of the KF.**

**Section 2.3.2**

- L 218: "R matrix is diagonal": for which observations is this true? What's the scale of weather patterns?

We are aware that the scale of synoptic weather patterns can be thousands of kilometers. Consequently, observation errors cannot be reasonably assumed to be uncorrelated over regions like Europe. However, this assumption is regularly used in GHG inversion studies, to simplify the inversion of the R matrix and because deriving the correlations between observation errors using statistical methods is challenging. If observation errors are correlated, several approaches can be employed to remove or mitigate the error correlations: (1) use another space of variables where the error covariance matrix R becomes diagonal, (2) average (temporally or spatially) the observations as often done for satellite observations, or (3) apply error inflation, as described in Chevallier et al. (2007). Additionally, observations with correlated errors can be processed using the batch EnSRF as an alternative. Based on this response, we added some information to the text.

Section 2.3.2: **If observation errors are correlated, several approaches can be employed to remove or mitigate the error correlations: (1) use another space of variables where the error covariance matrix R becomes diagonal, (2) average (temporally or spatially) the observations as often done for satellite observations, or (3) apply error inflation, as described in Chevallier et al. (2007). Additionally, observations with correlated errors can be processed using the batch EnSRF as an alternative.**

- P6, L160: The dimensions [n×N] should follow Z, not B, as B has dimensions [n×n].

Yes, indeed, it's a mistake. The square brackets should be near Z and not at the end of the sentence. Thanks a lot for spotting this! It has been corrected.

**Section 3.1.1**

- L 293: nlag 2: please note below this the reason for using nlag 2. Also maybe reference the results from Table 3. Maybe also the benchmark in Table D1.

We acknowledge that this information was missing. To address it, we added further details about using multiple lags at the beginning of Section 3.1 and included references to Table 3 and Table D1. We chose not to modify Section 3.1.1, as the values presented there serve only as an illustrative example. Although these values are identical to those used later in the experiments, we believe it is more appropriate to provide their justification in Section 4.2.1, where the synthetic experiments are introduced.

Beginning of section 3.1: **A larger number of lags 1) increases the computational cost but 2) may enhance the accuracy if emissions in the present window do not only affect the observations in the present but also in subsequent windows. These two statements are confirmed later by the synthetic experiments (see Table D1 and Table 3, respectively). One of the challenges in this inversion process is effectively managing the trade-off between the window length, the number of lags, and the computational cost.**

Section 4.2.1: **Over Europe, most of the air is flushed out of the domain within approximately 10–20 days. As a result, propagating information from local sources beyond 20 days into the future is unnecessary when performing inversions over Europe. However, observations at the beginning of a window are also sensitive to the emissions in the previous window, hence it is important to select at least two lags. We therefore select a window length of 10 days, providing three optimized values per month, and set a nlag value of 2 to balance computational efficiency and accuracy.**

- L 311: "eigenvalues and eigenvectors are already computed": how well does this scale? Is there a practical limit to the model resolution due to this inversion? Also maybe reference Table D1 (it shows that CIF performance is not the bottleneck).

We added a paragraph introducing the complexity of the eigenvalues computation and the generation of random vectors and how well they are scaling. As mentioned by the referee and as shown in Table D1, these steps are generally not the primary bottleneck in computational time, with CTM simulations being significantly more time-consuming. We have not reached a limit to the model resolution yet with our new code. However,

as generating the ensemble has a complexity of $O(n^2)$, doubling the resolution might become overwhelming in the future and we will maybe have to think about parallelizing further these specific operations.

Section 3.1.1: **The computation of eigenvalues is performed using the linalg.eigh function from the NumPy Python package, which has a computational complexity of $O(n^3)$. However, performing the decomposition of B via Kronecker products reduces this complexity to approximately $O(s^3)$, where s represents the number of variables to optimize within a single window. In contrast, the generation of random vectors has a complexity of $O(n^2)$ and can be computationally demanding at the start of the inversion, particularly for inversions spanning long periods. For typical real-data cases, such as a one-year inversion with a spatial resolution of approximately 0.25° over a domain like Europe, these two steps may take one to two hours to complete. However, as shown in Table D1, these steps are generally not the primary bottleneck in computational time, with model simulations being significantly more time-consuming.**

- L 338: it is not clear whether the ensemble in the runs in the publication is generated from the B matrix. Please note explicitly whether it is or why it is not.

We agree and we slightly modified the text to make it clearer.

Section 3.1.1: **In CIF, we employ the second method since the eigenvalues and eigenvectors of the B matrix are automatically computed when the YAML configuration file is read. Therefore, we first generate an ensemble of random vectors $z_i$ that each follow a multivariate Gaussian distribution $N(0, I_n)$ and then apply the formula $Q\Lambda^{1/2}Q^T z_i + x^b$ for each vector using Kronecker products to obtain random vectors that follow the distribution $N(x^b, B)$.**

**Section 3.1.2**

- L 356 to 358: this part is hard to understand. Suggestion: "for window 2 of the full period, this is the second time it is optimized from the observations."

Thanks for the suggestion, we applied it with slight modifications.

**Scaling factors corresponding to the second cycle are optimized using the observations in the third window. Note that the scaling factors of the second window are optimized for the second time, after having already been optimized in the first cycle using the observations from the second window.**

- Figure 1 is beautiful and clear!

Thanks a lot!

- L 361: why "nlag + 1"? The diagram only seems to show nlag.

Note that the posterior forward simulation must also be taken into account, leading to nlag + 1 simulations. For instance, for window 2, there is the prior 1, prior 2 and posterior 2, so three forward simulations spanning the window 2 in total. We modified the text.

**Each window is simulated nlag+1 times (nlag priors and one posterior).**

- L 362 to 363: this note may be better placed at the examples, 3.1.1 at line 294.

We agree and put this information rather at the beginning of Sect. 3.1.

Beginning of section 3.1: **A larger number of lags 1) increases the computational cost but 2) may enhance the accuracy if emissions in the present window do not only affect the observations in the present but also in subsequent windows. These two statements are confirmed later by the synthetic experiments (see Table D1 and Table 3, respectively). One of the challenges in this inversion process is effectively managing the trade-off between the window length, the number of lags, and the computational cost.**

**Section 3.2**

- L 405: "a wealth of" this is marketing language. Please give a short abstract of the features instead.

We agree and have decided to remove it. However, we prefer not to include an abstract here, as the features are described immediately afterward, making an abstract somewhat redundant.

**Section 3.2.1**

- L 417: is it possible to quantify how much was possible before and how big simulations could be used before and how big they can be now? Maybe rough, like "Factor X"?

We agree this is important and we added some rough estimation.

**CIF is now capable of easily performing complex operations for more than 500 samples in a few minutes, compared to roughly 50 samples before.**

**Section 3.3.3**

- eq 39 (line 479): if J(xa) is 0, this is 1. Are higher values better?

Yes, higher values means that the cost function have been better reduced. However, J(xa) cannot be equal to 0. There is necessarily a cost. We added some details.

**The optimal solution derived by the EnSRF minimizes the cost function defined in Eq. (5). To quantify this, we define the cost function reduction (CFR),**

**CFR = 1 − J(xa)/ J(xb)**

**A larger CFR indicates a greater reduction in the cost function.**

**Section 4**

In Section 4, a series of synthetic experiments for CO2 flux inversion were conducted to demonstrate the performance of the CIF-EnSRF method. In Appendix B, the authors conduct a further comparison with CTDAS, but this time they selected the synthetic inversion of CH4. There are some notable differences between CH4 and CO2 flux inversions; for example, the assimilation module for CH4 flux inversion involves generating separate perturbations for natural and anthropogenic emission categories from the B matrix, which is not the case for CO2. Could you clarify why CO2 flux inversion experiments were used earlier, but in Appendix B, CH4 flux inversion experiments were chosen for comparison instead of continuing with CO2 flux inversion?

We agree that this was unclear. Work on the present study began in the context of the EU Horizon 2020 project $CoCO_2$, a project dedicated to inverse modeling of $CO_2$. Before this work, ICON-ART inversions were conducted exclusively with CTDAS (Steiner et al. 2024), but those inversions were performed for $CH_4$. For practical reasons, we decided to perform the comparison for the same setup (with $CH_4$) as used in the study of Steiner et al. (2024).

These explanations have been incorporated into the paper.

Introduction: **This method therefore needed improvements, which were initiated when performing $CO_2$ inversions with CIF using different models and inversion setups as part of the Horizon 2020 $CoCO_2$ project (Berchet et al., 2023).**

Beginning of section 4: **Furthermore, we intend to identify elements that could have improved the initial real-data CIF-EnSRF inversions presented in Berchet et al. (2023), which were performed as part of the EU Horizon 2020 $CoCO_2$ project. For this purpose, we maintain consistency by performing $CO_2$ inversions and using the same input data.**

Appendix B: **Before this work, ICON-ART inversions were conducted exclusively with CTDAS (Steiner et al., 2024), but those inversions were performed only for $CH_4$. As mentioned in Sect. 3.2.2, it is straightforward to switch from one species to another in CIF. We therefore selected the reference synthetic inversion (case 1) of $CH_4$ fluxes over Europe performed by Steiner et al. (2024) with CTDAS-ICON-ART and aimed to replicate it precisely with CIF-ICON-ART. The complete setup of the CTDAS-ICON-ART inversion is detailed in Steiner et al. (2024), hence we only outline the main components of the configuration here:**

**...**

**We applied the exact same setup for the CIF inversions. Additionally, the same ensemble (i.e., same ensemble size and same sample values) was employed in both inversions to eliminate the influence of randomness. We also endeavored to replicate the post-processing of ICON outputs (i.e., time, horizontal and vertical interpolations) as closely as possible in CIF.**

[Figure]

**Section 4.1.1**

- line 569: are higher resolution possible with this implementation? For example 6.5km? This might significantly improve the possible gains from high frequency column measurements. Are there boundary conditions for the Europe area?

We have already successfully tested a resolution of 13 km and we will test a resolution of 6.5 km with the new GPU code of ICON-ART in a few months. We can think of four potential major bottlenecks if the size of the control vector is too high: 1) the eigenvalues and eigenvectors will take too much time to be calculated, 2) the ensemble will take too much time to be generated if there are any spatial correlations to consider, 3) the memory of the compute node will be overwhelmed before the number of samples in the ensemble is high enough or/and 4) the post-processing of model output data will take too much time. However, we believe none of these bottlenecks will appear at a resolution of 6.5 km, especially on supercomputers with large memory and powerful computed nodes such as the new Alps system at the Swiss supercomputing center CSCS. Furthermore, it is also still possible to reduce $n$ by optimizing emissions for batches of cells, rather than for individual cells.

Boundary conditions for Europe are usually provided by the CAMS products for $CO_2$, $CH_4$ and $N_2O$, which are derived from global inversions at coarse resolutions (see Sect. 4.1.2). The boundary conditions could also be derived directly from the measurements.

We added some information in the text.

Section 4.1.1: **Although a coarse resolution is used here to demonstrate the new system and conduct numerous sensitivity tests, finer horizontal resolutions, up to 13 km, have already been successfully tested with ICON-ART.**

**Section 4.1.3**

P24, Figures 3a and 3b show the map of the true and posterior scaling factors. However, I have a question: why not include a comparison of the maps for the prior, posterior, and true fluxes, as flux maps could provide a more direct view of their spatial distribution? Additionally, are the values of the prior scaling factors equal to 1?

We chose to present the scaling factors in the paper because they are much easier to compare than the fluxes. Below, we provide the prior, posterior, and true fluxes so the referee can form his/her own opinion. With scaling factors, we can more easily see the modification patterns and the spatial correlations (batches of colors).

Additionally, presenting the fluxes would require five subplots instead of four, which we believe would make the figure less clear and more cluttered. This choice prioritizes clarity and practicality in the presentation. We have decided to include the figure below as supplementary figure (Figure D2) in the new version so readers can also access the fluxes themselves.

And yes, the values of the prior scaling factors are all equal to 1 here. But prior values can be selected by users in CIF. We have slightly modified the text in Section 4.1.3 to make it clearer.

Perturbed fluxes representing the truth are then obtained by applying these scaling factors to the respiration fluxes, while keeping other fluxes unperturbed (i.e. scaling factors are equal to 1).

**Figure D2.** Prior, posterior and true respiration $CO_2$ fluxes from the N200L600 experiment, for which the ER and MER computations are presented in Figure 3. Prior and true fluxes are identical for all synthetic experiments.

**Section 4.2.1**

- Figure 3: please note that this is the best result from the LEVEL 1 comparison.

No, the best result is not necessarily N200L600 for all countries. N300L600 can show better MER, as shown in Figure 4 and new Table D3. As explained in Sect. 4.2.2, an ensemble size of 200 is typically employed for running inversions with ICON-ART, in order to balance computational cost and inversion quality (Steiner et al., 2024). As the best LEVEL1 results with this ensemble size are obtained with a localization length of 600 km, N200L600

is adopted as the control experiment for all families. We slightly modified the caption of Figure 3 to explain why N200L600 is shown here.

Figure 3: **Computation of error reduction for N200L600, which is adopted as the control experiment for all families in LEVEL2 experiments.**

- Figure 3: please add more explanation to these results. Why are the neighboring cells so different in the error reduction (c)? Also please provide a table with the values of d (the clearest result) so readers can quantitatively compare the error reduction per country.

We agree that the analysis of Figure 3 needed improvement. To address this, we added a detailed paragraph in Section 4.3 focusing on Figure 3 and explaining the reasons behind the significant differences between neighboring cells. Additionally, as requested, we included Table D3, which provides the MER for all EU27 countries, as well as Switzerland, the UK, and Norway.

Section 4.3: **Figure 3 illustrates the process of calculating the ER and MER for each country in Europe, based on the true and posterior scaling factors, and the prior fluxes. The ER can exhibit strong spatial heterogeneity for two main reasons. First, the spatial distribution of posterior scaling factors is generally smoother than that of the true scaling factors because the constraints provided by surface observations are insufficient to fully capture the spatial variability of the true scaling factors. Second, when fluxes within a region are spatially non-uniform, the system has difficulty distinguishing low-flux cells from high-flux cells. This limitation can result in large relative errors for cells with low fluxes. The MER reduces the influence of errors associated with low fluxes, providing a reliable estimate of how accurately the cells within a country can be evaluated by the system.**

Section 3.3.1: **Figure 3 illustrates an example of MER computation over Europe based on a set of prior, posterior and true scaling factors.**

**Section 4.2.2**

- L 641: reference the results of this experiment in 3.1.1 to motivate why nlag 2 is used in the example.
- Table 2: why is the maximum lag 3? Would lag 5 be possible? Which lags are used in current CTDAS runs? Reference Table D1.

The experiment in Sect. 3.1.1 is intended solely as an illustrative example. While we could have used nlag=5, it was simpler to set nlag=2 for the purpose of creating Figure 1. In response to one of the referee's earlier comments, we already updated Sect. 4.2.1 to explain the rationale for using two lags and why using more than three lags would be useless if the window length is set to 10 days. The number of lags in CTDAS (and CIF) really depends on the application. For global $CO_2$ inversions, the number of lags can be 12 (Peters et al., 2005). For regional applications, the number of lags and the window length should be selected so the total length of a cycle does not exceed the typical residence time of air inside the domain.

Note that Table D1 provides the amount of CPU hours used to perform LEVEL1 experiments and therefore has no relevance for Table 2, which describes LEVEL2 experiments. However, it has relevance for Table 1, which describes LEVEL1 experiments. We therefore included it in the caption of Table 1.

Section 4.2.1: **In particular, we use a window length of 10 days, with two lags, resulting in five cycles of 20 days each. Over Europe, most of the air is flushed out of the domain within approximately 20–30 days. As a result, propagating information from local sources beyond 30 days into the future is unnecessary when performing inversions over Europe, but it is important to allow these sources to influence observations for at least 20 days. To achieve this, we select a window length of 10 days, providing three optimized values per month, and set a nlag value of 2 to balance computational efficiency and accuracy. The sensitivity to the number of lags is tested in LEVEL2 experiments.**

Table 1: **Table D1 provides the amount of CPU hours used to perform these experiments.**

Section 4.2.2 and Table 2: In LEVEL2 experiment (F), the authors investigated the impact of adjusting the mean and variance of the ensemble on the inversion results. Please provide the values for the mean and variance of the ensemble in both the adjusted and unadjusted experiments, i.e., what were the mean and variance values used in Table 2 for EXP_F_f1* and EXP_F_t1*? Only by providing the specific data parameters and comparing the extent of their changes can the importance of this parameter be better demonstrated.

We added the values to Table 2.

Members of the ensemble (samples) are stored in a 2-dimensional array of dimension (N, n).  The first dimension of the array is the index of the sample (size N) and the second dimension is the index of the scaling factor/optimized variable (size n). In other words, there is one value per sample and per optimized variable. Adjusting the mean and the variance means that each scaling factor should have a mean of 1 and a variance of 1 over the N samples after the adjustment.

In Table 2, we show the mean ± std of the two distributions represented by 1) the 1-d dimensional array (of dimension n) filled with the means calculated across the N samples and 2) the 1-d dimensional array (of dimension n) filled with the variances calculated across the N samples, respectively.

We added these explanations to the caption of Table 2 and included the values in Table 2. We also removed the star symbols that we forgot to remove before submitting the manuscript.

**Table 2. Description of LEVEL2 experiments. Apart from the parameters described in this table, the LEVEL2 experiments all share the same configuration. [1] For each of the *n* optimized variables, an average across the N samples is calculated. A distribution of ensemble averages is therefore created. The values presented here represent the mean and standard deviation computed over this distribution. [2] Same as before but the distribution is made with the variance over the N samples, for each optimized variables.**

| EXP_F_f1 | 50 | 600 | Adjusting the mean and variance of the ensemble = False |
| | | | [1]Means = 1.00 ± 0.15 / [2]Variances = 0.98 ± 0.20 |

| | | | |
|---|---|---|---|
| **EXP_F_t1** | **50** | **600** | **Adjusting the mean and variance of the ensemble = True**
**[1]Means = 1.00 ± 0.00 / [2]Variances = 1.00 ± 0.00** |
| **EXP_F_f2** | **200** | **600** | **Adjusting the mean and variance of the ensemble = False**
**[1]Means = 1.00 ± 0.07 / [2]Variances = 1.00 ± 0.10** |
| **EXP_F_t2** | **200** | **600** | **Adjusting the mean and variance of the ensemble = True**
**[1]Means = 1.00 ± 0.00 / [2]Variances = 1.00 ± 0.00** |

**Section 4.3**

- Figure 5, d: why does this show France, when the maps in Figure 4 show little change in MUR in France? It could be good to split out MUR from Figure 5 and add a set of plots for MUR in different countries. And give possible explanations of the differences. Portugal has no direct observations, but changes. MUR in UK also seems to change much more than in France. There are also countries in eastern Europe with bigger changes. Are these effects from being at the border of the simulated area?

- Line 693: as above, see the maps in Figure 4. Other countries seem to show bigger changes than France.

We address both comments in this response.

It seems the referee may have confused the Mean Uncertainty Reduction (MUR) with the Mean Error Reduction (MER), which is the only metric shown in Figure 4. The MUR quantifies the reduction in posterior uncertainty relative to prior uncertainty, whereas the MER measures how closely the posterior emissions align with the true emissions. Importantly, the MUR is applicable to both synthetic and real-case experiments, while the MER applies only to synthetic experiments. Figure 4 presents the MER (both domain-total and country-totals), whereas Figure 5d displays the MUR. Therefore, we do not believe it would be relevant to split Figure 5d by country, as including the MUR for every country would not add meaningful insights. As described in Sect. 4.3, we highlight France in Figure 5d because, among the countries within the observation network, its metrics are most affected by changes in the number of samples and localization length. Specifically, the MUR over France clearly illustrates how the EnSRF tends to become increasingly overconfident in its solutions as the number of samples decreases.

To provide additional clarity, we added Table D3, which presents the MER for most European countries and corresponds to the data shown in Figure 4. This table highlights that, among countries with an MER of at least 20% in N200L600 (mainly those covered by or close to the observation network), France exhibits the largest change between N50Lnone and N200L600 (+68%). For example, the MER in the UK rises from 6% in N50Lnone to 63% in N200L600 (+57%).

For countries poorly covered by the observation network, randomness can significantly influence results. When a low number of samples and no localization are used, the results are dominated by noise, which explains the very poor or occasionally high MER values in some European countries. As shown in our experiments, increasing the number of samples reduces the effect of randomness, while localization further stabilizes results. However, localization can also exclude relevant information for countries located far from the observations, as discussed in Sect. 4.3. This is why countries at the domain's edge may show extreme differences, ranging from very low MER values (due to noise) to a MER=0, indicating no changes (here meaning no information retrieval) for these countries.

We revised the text to enhance clarity and included additional details to explain why the MER varies for countries at the domain boundaries.

**Countries near observing sites, such as those in Western and Central Europe, benefit from a reduced localization length, regardless of the number of samples. However, when the number of samples is reasonable, decreasing the localization length below a certain threshold can start filtering out relevant information. This effect is evident in countries farther from observation sites, such as Portugal, Spain, and those in the Balkans or Eastern Europe. For these countries, the MER (whether initially positive or negative without localization) tends toward 0 % as the localization length decreases. This indicates not only the loss of potential meaningful information but also the suppression of any problematic effects from random noise.**

**...**

**We highlight France here because, among the countries covered by the observation network, its metrics are the most affected by changes in the number of samples and localization length.**

- line 751: the note about the heaviside function seems to be evading a result. Maybe compare with the other functions at larger correlation length?

We rather decided to remove the last two sentences of this paragraph, as there are already four sensitivity tests for this family. Doubling this number would place disproportionate emphasis on this topic, which is not of critical importance.

- line 760: "to the best of our knowledge" sounds like this sentence should not be stated. Remove?

Yes we agree. We removed this sentence.

**Section 4.5**

- line 777: "consistent with established literature": please reference appendix B.

We added a reference to appendix B in the corresponding sentence.

**The overarching aim of these experiments (including those presented in Appendix B) was to validate that the CIF-EnSRF system produces results consistent with established literature.**

[Figure]

**Appendix B**

P37, L844-L858: What are the differences between the CTDAS configuration and the CIF-EnSRF configuration here? It would be helpful to provide a comparison of their configurations and explain the similarities and differences.

For the inversions presented in Appendix B, the inversion setups are identical (see the list of setup elements in Appendix B). As noted, the minor differences observed are likely due to undetected discrepancies in the post-processing of ICON outputs. We revised the text to clarify this point.

Appendix B: **To provide a robust comparison, we needed to perform two inversions (one with CTDAS and one with CIF) with identical setups, data processing, and assimilation methods (e.g., the same assimilation order). Before this work, ICON-ART inversions were conducted exclusively with CTDAS (Steiner et al., 2024), but those inversions were performed only for $CH_4$. As mentioned in Sect. 3.2.2, it is straightforward to switch from one species to another in CIF. We therefore selected the reference synthetic inversion (case 1) of $CH_4$ fluxes over Europe performed by Steiner et al. (2024) with CTDAS-ICON-ART and aimed to replicate it precisely with CIF-ICON-ART. The complete setup of the CTDAS-ICON-ART inversion is detailed in Steiner et al. (2024), hence we only outline the main components of the configuration here:**

**...**

**We applied the exact same setup for the CIF inversions. Additionally, the same ensemble (i.e., same ensemble size and same sample values) was employed in both inversions to eliminate the influence of randomness. We also endeavored to replicate the post-processing of ICON outputs (i.e., time, horizontal and vertical interpolations) as closely as possible in CIF.**

**Appendix D**

- Table D1: please reference this table from table 2, page 26. Please give some additional information, how this benchmark data was gathered.

Note that Table D1 provides the amount of CPU hours used to perform LEVEL1 experiments and therefore has no relevance for Table 2, which describes LEVEL2 experiments. However, it has relevance for Table 1, which describes LEVEL1 experiments. We therefore included it in the caption of Table 1.

We also included details on how we separately retrieved the CPU hours consumed by CIF and ICON.

Table 1: **Table D1 provides the amount of CPU hours used to perform these experiments.**

Table D1: **At the end of each job performed on the Piz Daint supercomputer, the logs provide the total CPU hours used. To execute the inversion, a parent job running CIF periodically initiates sub-jobs to perform ICON simulations. It allows us to track the CPU hours consumed separately by CIF and ICON.**

[Figure]

[Figure]

**Technical comments**

- L 18: add a comma: (N2O), or

Thanks for spotting this. It has been modified.

- L 231: the note duplicates parts of line 218. Please disentangle the two sentences to avoid the duplication.

We agree this is redundant. We changed this.

Beginning of section 2.3.2: **In this case, batch EnSRF and serial EnSRF are mathematically equivalent (Kotsuki et al., 2017; Nerger, 2015; Whitaker and Hamill, 2002) and thus provide identical results.**

- L 249: wrong hyphenation: anal-ysis — might need explicit hyphenation via \hyphenation{a-na-ly-sis}

Thanks for the tip. We applied this.

- L 283: which format is used? Where is it described? Citation, or a footnote with a link?

We added this information.

**Through the YAML configuration file of CIF (http://yaml.org, last access: 12 December 2024), users can define fundamental settings for the inversion process:**

- L 314 to 122: a long discussion for a simple point. Suggestion: "the default behavior of CIF is to erase the scaling factors after conversion. But some models (including ICON-ART) need scaling factors as import, so they are preserved here." Feel free to shorten further.

We shortened it.

**The default behavior of CIF is to erase scaling factors after conversion. However, certain models, such as ICON-ART, require scaling factors instead of perturbed fluxes. The additional sample allows to retrieve the former from the latter.**

- L 333: "and a scaling" this seems to be missing the word "factor".

Correct. We added "operation" to avoid a confusion with the scaling factors of the ensemble.

- line 635: please add a note that the results are discussed in section 4.3.

We added a note at the end of Sect. 4.2.1 for LEVEL1 experiments and at the end of Sect 4.2.2 for LEVEL2 experiments.

**Appendix B**

- Figure B1: please label the colorbars (scaling factor / unitless)

This has been done.

- Figure B2: please label the colorbars with the variable shown (uncertainty reduction / unitless)

This has been done.

**Appendix C**

- Figure C1: please show the lines with different line styles to be recognizable in grayscale.

This has been done.

[Figure]

**REFERENCES**

Berchet, A., Thanwerdas, J., Reum, F., Elias, E., and Broquet, G.: D5.3 Quantification of transport errors and database of optimized fluxes and simulations for an ensemble of models and inversion set-up | CoCO$_2$: Prototype system for a Copernicus CO$_2$ service, https://coco2-project.eu/sites/default/files/2023-11/CoCO2-D5-3-V0.1.pdf, 2023.

Chevallier, F.: Impact of correlated observation errors on inverted CO2 surface fluxes from OCO measurements, Geophysical Research Letters, 34, https://doi.org/10.1029/2007GL030463, 2007.

Peters, W., Miller, J. B., Whitaker, J., Denning, A. S., Hirsch, A., Krol, M. C., Zupanski, D., Bruhwiler, L., and Tans, P. P.: An ensemble data assimilation system to estimate CO$_2$ surface fluxes from atmospheric trace gas observations, Journal of Geophysical Research: Atmospheres, 110, https://doi.org/10.1029/2005JD006157, 2005.

Rieger, D., Bangert, M., Bischoff-Gauss, I., Förstner, J., Lundgren, K., Reinert, D., Schröter, J., Vogel, H., Zängl, G., Ruhnke, R., and Vogel, B.: ICON–ART 1.0 – a new online-coupled model system from the global to regional scale, Geoscientific Model Development, 8, 1659–1676, https://doi.org/10.5194/gmd-8-1659-2015, 2015.

Steiner, M., Peters, W., Luijkx, I., Henne, S., Chen, H., Hammer, S., and Brunner, D.: European CH$_4$ inversions with ICON-ART coupled to the CarbonTracker Data Assimilation Shell, Atmospheric Chemistry and Physics, 24, 2759–2782, https://doi.org/10.5194/acp-24-2759-2024, 2024.

Zängl, G., Reinert, D., Rípodas, P., and Baldauf, M.: The ICON (ICOsahedral Non-hydrostatic) modelling framework of DWD and MPI-M: Description of the non-hydrostatic dynamical core, Quarterly Journal of the Royal Meteorological Society, 141, 563–579,1295 https://doi.org/10.1002/qj.2378, 2015